# Towards General Modality Translation with Contrastive and Predictive Latent Diffusion Bridge

**Nimrod Berman** [1,2] *  **Omkar Joglekar** [1,3] *  **Eitan Kosman**[1]  **Dotan Di Castro**[1]  **Omri Azencot**[2]

[1]Bosch AI Center  [2]Ben-Gurion University of the Negev  [3]Technical University of Munich

## Abstract

Recent advances in generative modeling have positioned diffusion models as state-of-the-art tools for sampling from complex data distributions. While these models have shown remarkable success across single-modality domains such as images and audio, extending their capabilities to *Modality Translation (MT)*, translating information across different sensory modalities, remains an open challenge. Existing approaches often rely on restrictive assumptions, including shared dimensionality, Gaussian source priors, and modality-specific architectures, which limit their generality and theoretical grounding. In this work, we propose the Latent Denoising Diffusion Bridge Model (LDDBM), a general-purpose framework for modality translation based on a latent-variable extension of Denoising Diffusion Bridge Models. By operating in a shared latent space, our method learns a bridge between arbitrary modalities without requiring aligned dimensions. We introduce a contrastive alignment loss to enforce semantic consistency between paired samples and design a domain-agnostic encoder-decoder architecture tailored for noise prediction in latent space. Additionally, we propose a predictive loss to guide training toward accurate cross-domain translation and explore several training strategies to improve stability. Our approach supports arbitrary modality pairs and performs strongly on diverse MT tasks, including multi-view to 3D shape generation, image super-resolution, and multi-view scene synthesis. Comprehensive experiments and ablations validate the effectiveness of our framework, establishing a new strong baseline in general modality translation. For more information, see our project page: `https://sites.google.com/view/lddbm/home`.

## 1 Introduction

Generative modeling has progressed rapidly in recent years, with diffusion models emerging as a powerful framework for sampling from complex, high-dimensional distributions [57]. These models have achieved impressive results across a range of domains, including images, audio, and video [30, 9, 21]. Their inherent conditioning module naturally lends itself to the next step: *Modality Translation (MT)*, the task of mapping information across different sensory modalities, such as from text to image or from audio to video [72, 68, 6, 2]. Unlike traditional generation tasks, MT often involves bridging modalities that differ significantly in dimensionality, structure, and semantics, posing unique challenges for modeling shared representations and consistent cross-modal synthesis.

Recent works on distribution translation, notably Denoising Diffusion Bridge Models (DDBMs) [75], offer a principled framework for modeling transitions between arbitrary distributions. In parallel, diffusion models have been widely applied to MT tasks such as text-to-image [49] and text-to-3D generation [46, 56], typically by conditioning on auxiliary signals and exploiting modality-specific inductive biases. However, these approaches often assume translation between distributions of similar dimensionality or insert strong data biases into their design. This limits their applicability to more general MT scenarios involving heterogeneous modalities, e.g., translating from images to 3D

---

*Equal contribution

structure or from low-resolution quality to a high-resolution one. Moreover, many implementations rely on neural architectures tailored to specific data types, such as U-Nets [52], which are well-suited to grid-based data but struggle with abstract or unstructured modalities. These limitations highlight the need for a general framework that minimizes reliance on modality-specific design and remains theoretically grounded across a broad range of cross-modal tasks.

Latent Diffusion Bridges such as [24, 36, 29, 25, 70] attempt to bridge the gap between diffusion models and general modality translation. However, most existing works adopt this formulation primarily for computational reasons—either to reduce the cost of the diffusion process itself [50, 75], or to improve sampling efficiency through direct endpoint mappings [24] and cannot be applied to general MT purposes. In order to take a step towards a general effective framework, we suggest a new latent extension of DDBMs that facilitates generality for MT tasks along with effective performance.

*LDDBM* extends DDBMs to a shared latent space and learns a bridge between embeddings of disparate modalities, enabling translation without requiring the two domains to share dimensionality. Concretely (see Fig. 2), we: *(i)* encode source and target examples into a common latent space using simple modality-specific encoders; *(ii)* apply a *latent diffusion bridge* implemented with a Transformer denoiser that uses cross-attention to condition on the source latent while predicting the target latent; and *(iii)* decode the predicted latent back to the target modality. To promote semantic consistency, we utilize paired training data with a *contrastive alignment loss* (inspired by CLIP [48]) that pulls corresponding pairs together and pushes unrelated pairs apart. To ensure the bridge performs end-to-end translation, and not just local denoising, we add a *predictive loss* that compares the final decoded output to the ground-truth target. Finally, we reduce architectural bias with a domain-agnostic denoiser and use a simple yet effective *iterative training scheme* that alternates between alignment and denoising steps, improving stability and performance.

This approach offers several key advantages. First, it supports arbitrary modality pairs without relying on fixed priors, shared dimensionality constraints, or hand-crafted architectures—aside from the encoder and decoder used to project data into the latent space. Second, the contrastive loss enhances semantic coherence across domains, and our bridge architecture achieves strong performance on a range of tasks, including multi-view to 3D shape generation [55], super-resolution [38], and scene generation from multi-view cameras [62]. Additionally, we provide a simple, flexible, and theoretically grounded framework for MT tasks and demonstrate its state-of-the-art performance. Finally, we conduct a thorough ablation study to evaluate each component of our framework, identifying the key elements that contribute to its improved performance.

## 2    Related Work

**Diffusion models** [57, 19] and score-based generative models [22] have emerged as powerful approaches for wide range of modalities generation [4, 42, 67, 69, 5, 14, 12], often surpassing the performance of GANs [15], which previously defined the state-of-the-art. A key strength of diffusion models lies in their inherently conditional architecture, which enables injecting of auxiliary signals such as diffusion time steps and text embeddings. This architectural flexibility has positioned diffusion models as a natural fit for general Modality Translation (MT) tasks. Advancements in sampling techniques [19, 58, 27], particularly innovations like classifier-free guidance [20, 9], have further enhanced the generative capabilities of these models in diverse MT scenarios. While these developments are broadly applicable and do not rely on domain-specific inductive biases, several works have leveraged the conditional structure of diffusion models with tailored data-specific adaptations to excel in particular MT tasks [21, 46, 38, 66]. Since our objective is general-purpose modality translation, our method is best compared to general diffusion-based approaches that maintain broad applicability across domains [27, 35].

**Distribution Translation.** While diffusion models are among the most effective tools for general modality translation (MT) tasks, Denoising Diffusion Bridge Models (DDBMs) have shown strong performance for translating between modalities within the same data domain [75]. This framework extends the standard denoising diffusion paradigm [19] from noise-to-data generation to enabling data-to-data translation between paired distributions. Other principled approaches, such as Optimal Transport (OT) [1] and Schrödinger bridges [59, 32, 54], offer alternative ways to model transitions between arbitrary distributions. Diffusion Bridge Implicit Models (DBIMs) [73] generalize DDBMs using non-Markovian bridges over discretized timesteps, reducing resource demands while preserving marginal distributions and training objectives. Similarly, Consistency Diffusion Bridge Models

(CDBMs) [16] improve efficiency by learning the consistency function of the probability-flow ODE (PF-ODE), enabling direct prediction at arbitrary time points. Recent work [63] further mitigates PF-ODE singularity issues at the reverse-time initialization via posterior sampling, improving continuity and reducing discretization error. Despite this progress, DDBMs and their variants remain limited to shared-modality settings and are not directly applicable to general MT tasks. Other frameworks, such as CLIP [48], diffusion-based models [19, 58], and multimodal models [53, 3, 13, 76], leverage shared embeddings or conditional generation to bridge modalities. However, end-to-end modality-to-modality translation remains underexplored and lacks unified methodologies. While unpaired translation [77] and weakly paired setups [37] have received substantial attention, our work, as well as the models above, operates within the paired translation paradigm [23].

Finally, we note several concurrent efforts that share conceptual similarities but differ substantially in scope and formulation. CrossFlow [33] and FlowTok [17] both employ flow-based formulations to map directly between text and image modalities. While effective within that specific domain, they rely on task-specific design choices (e.g., text-image tokenization, contrastive language supervision) and thus do not generalize to arbitrary modality pairs. In contrast, our work pursues a *general-purpose latent bridging framework* that supports heterogeneous modalities, including 3D and audio, and is not tied to any particular data type or dimensionality. Moreover, the underlying formulations differ: the above methods are based on *Flow Matching*, whereas our approach extends the *Denoising Diffusion Bridge Model* (DDBM) framework, leading to distinct training objectives and sampling dynamics. Another concurrent work, DPBridge [24], adopts a latent diffusion bridge for dense prediction but remains limited to image-to-image translation, leveraging a pre-trained Stable Diffusion backbone. Unlike DPBridge, our method introduces contrastive and predictive objectives for alignment, a modality-agnostic architecture trained from scratch, and comprehensive studies on training strategies, enabling translation across diverse modality pairs beyond images.

## 3 Background

**Diffusion Models** [57] have emerged as leading generative models, achieving state-of-the-art performance across various tasks [19, 30]. Intuitively, these models operate by mapping a latent representation, sampled from a simple prior distribution (e.g., Gaussian), through an iterative process to meaningful outputs such as images or audio. However, their confinement to transformations that originate from a predefined simple distribution poses a fundamental limitation, as it restricts the ability of diffusion models to translate between arbitrary and potentially complex data distributions flexibly. Researchers have proposed several approaches to address the challenge of translating between arbitrary distributions. This work focuses on DDBMs [75], which extend diffusion models to connect endpoints from two arbitrary image distributions of the same dimensionality.

**Denoising Diffusion Bridge Models** extend diffusion models to enable transformations between two arbitrary distributions by conditioning the diffusion process on fixed, known endpoints using the Doob's $h$-transform [10]. Given two distributions $p(x_0), p(x_T)$ and for a given joint distribution $p(x_0, x_T)$, where the subscripts denote the time associated with a state, the Doob's transform allows the diffusion process to be initialized at a specific state $x_0$ at time 0 and terminate at a desired state $x_T$ at time $T$. Its associated forward stochastic differential equation reads,

$$\mathrm{d}\,x_t = f(x_t, t)\,\mathrm{d}\,t + g(t)^2 h(x_t, t, x_T, T)\,\mathrm{d}\,t + g(t)\,\mathrm{d}\,w_t \,, \tag{1}$$

where the functions $f, h, g$ can be calculated analytically, similar to adding noise with a particular scheduler in standard diffusion models. DDBM aims to learn the reverse process, which is given by

$$\mathrm{d}\,x_t = \left[f(x_t, t) - g^2(t)\left(\nabla_{x_t} \log q(x_t \mid x_T) - h(x_t, t, x_T, T)\right)\right]\mathrm{d}\,t + g(t)\,\mathrm{d}\,\hat{w}_t \,, \tag{2}$$

where $\hat{w}_t$ is the Wiener process in reverse time and $\nabla_{x_t} \log q(x_t \mid x_T)$ is known as the score function. Let $s_\theta(x_t, x_T, t)$ denote a neural network approximation of the score function. To train the above model, we consider the following loss objective,

$$\mathbb{E}_{x_t, x_0, x_T, t}\, w(t)\left|s_\theta(x_t, x_T, t) - \nabla_{x_t} \log q(x_t \mid x_0, x_T)\right|^2 \,. \tag{3}$$

In practice, we utilize the variance exploding (VE) scheme in this work from [75] where $\nabla_{x_t} \log q(x_t \mid x_0, x_T) = (x_T - x_t)/(\sigma_T^2 - \sigma_t^2)$ and $x_t \sim \mathcal{N}(x_0, \sigma_t^2 I)$.

Finally, a process can be formulated to approximately sample from $p(x_0 \mid x_T)$ by reversing the diffusion bridge. Using a training set of paired samples drawn from $p(x_0, x_T)$, it is possible to

learn how to map states from an arbitrary distribution $p(x_T)$ to another arbitrary distribution $p(x_0)$. However, DDBMs rely on the explicit assumption that both $p(x_0)$ and $p(x_T)$ reside in the same space $\mathbb{R}^d$ for some $d$. Consequently, this framework, and similar approaches, are not applicable to our multi-modal setup, where paired data do not share the same dimensionality or underlying manifold. To overcome this challenge of translating between distributions of different dimensionality, we connect representations of the two endpoints achieved through encoders and decoders that can be potentially pre-trained, which bring the endpoints to the same dimensionality.

## 4  Method

In this section, we first formulate the problem of task-agnostic translation. Then, we gradually present our approach to solving task-agnostic translation, depicted in Figure 2, beginning with introducing the framework and objectives and then moving to improved and adapted objectives and model architecture. Finally, we discuss the training procedure to achieve optimal translation performance.

### 4.1  Problem Formulation

We study the general task of translation between two modalities. Although researchers have extensively explored this task across various domains [46, 56], existing methods typically tailor their solutions to specific tasks and depend heavily on architectural biases or strong supervision. In contrast, we propose a task-agnostic framework capable of translating between any two modalities represented by shared multimodal distributions $p(x)$ and $p(y)$, where a joint distribution $p(x, y)$ exists. Given a target modality distribution $p(x)$, where $x \in \mathbb{R}^k$, and a source modality distribution $p(y)$, where $y \in \mathbb{R}^s$ with $k \neq s$, along with a dataset of paired samples $(x, y) \sim p_{\text{data}}(x, y)$, the goal of multimodal translation is to learn a model $M$ that approximates the conditional distribution $p(x \mid y)$ or $p(y \mid x)$, thereby enabling translation from one modality to the other.

### 4.2  Multimodal Translation Framework

While dimensional mismatches between modalities can sometimes be solved via simple heuristics, for example, padding a smaller image to match the size of a larger one, these solutions generally under-perform, and many tasks, such as translating 2D to 3D data, are inherently more complex. To tackle this, we introduce a latent structure that bridges the modalities, providing a theoretical and practical framework for addressing this gap. Although related ideas have been explored in the context of computational reductions [75] and specific tasks [33, 17], to the best of our knowledge, they have not been investigated in the setting of general multimodal translation. We model the translation process via a *probabilistic latent bridge* to realize this idea. We aim to estimate the conditional distribution $p(x \mid y)$; however, direct modeling of this distribution is often intractable. To address this, we introduce intermediate latent variables and associated distributions: $p(z_0 \mid x)$ and $p(z_T \mid y)$, where $z_0, z_T \in \mathbb{R}^d$ reside in a shared latent space. In settings where reconstruction of $x$ is required, we additionally define a decoder distribution $q(x \mid z_0)$. The core objective is thus re-framed as modeling the conditional $p(x \mid y)$ through the composition: $p(x|y) = p(z_T|y) \, p(z_0|z_T) \, q(x|z_0)$. Here, $p(z_T \mid y)$ and $q(x \mid z_0)$ serve as the encoder $E_y$ and decoder $D_x$ and are implemented as modality-specific neural networks (e.g., MLPs or CNNs). During training, the encoding of $x$ through the encoder $E_x$ ($p(z_0 \mid x)$) is also required to compute the bridge loss, but this is not needed during inference. The bridge $p(z_0 \mid z_T)$ is modeled using a DDBM module, although our framework is agnostic to this choice and can be extended to other choices such as those proposed in [1, 19, 27, 35]. The choice of encoder and decoder is the only modality-specific component in our framework. In our experiments, we employed either very simple neural architectures or pre-trained models.

### 4.3  Basic Latent Bridge Loss

Our multimodal translation objective comprises two loss components: an autoencoder loss and a diffusion bridge loss. The autoencoder reconstruction loss penalizes poor reconstruction of inputs. Let $E_x$ and $D_x$ denote the encoder and decoder, respectively, of an autoencoder $\text{AE}_x$ for the $x$ modality, i.e., $\text{AE}_x(x) := D_x \circ E_x(x)$, then the **autoencoder loss** is given by

$$\mathcal{L}_{\text{AE}_x} = d(x, \text{AE}_x(x)) \,, \tag{4}$$

where $d$ is a distance metric such as $\ell_2$ or LPIPS [71]. A similar objective can be formulated for the $y$ modality. To train the latent diffusion bridge, we adopt the score matching **bridge loss** from the DDBM formulation, applied on the latent variables,

$$\mathcal{L}_{\text{bridge}} = \mathbb{E}_{z_t, z_0, z_T, t} \left[ w(t) \left| s_\theta(z_t, z_T, t) - \nabla_{z_t} \log q(z_t | z_0, z_T) \right|^2 \right] \,, \tag{5}$$

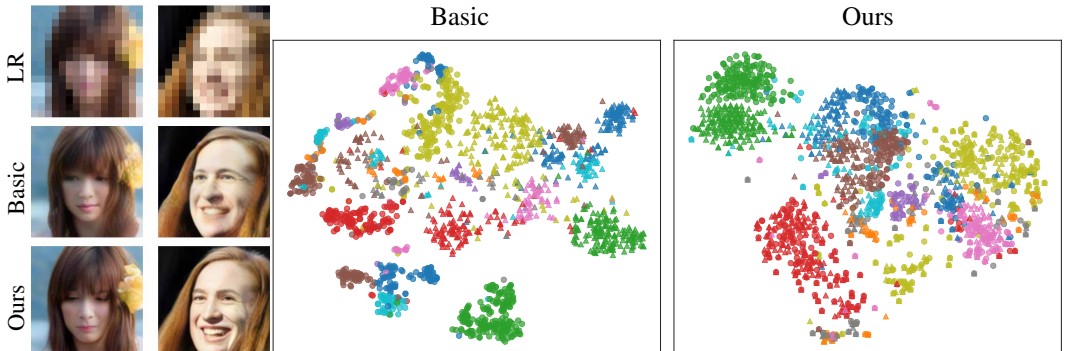

Figure 1: Left: Super-resolution examples obtained with the basic approach vs. ours. Right: Latent space structure of the basic formulation vs. our latent space.

where $z_t$ is a noisy version of $z_0 := E_x(x)$ at timestep $t$, $w(t)$ is a weighting function over time, and $z_T := E_y(y)$. Combined together, we obtain the following training objective

$$\mathcal{L} = \mathcal{L}_{\text{bridge}} + \mathcal{L}_{\text{AE}_x} + \mathcal{L}_{\text{AE}_y} \ . \tag{6}$$

This objective can be optimized in two phases (reconstruction followed by bridge training) or jointly in an end-to-end fashion. Next, we show that this basic setup, while effective, is insufficient from several different perspectives.

### 4.4 Our Bridge Approach

Our baseline loss formulation consists of three components: a standard score matching loss applied to the diffusion bridge and two separate reconstruction losses—one for the source distribution and one for the target, each responsible for mapping data to its respective latent space and reconstructing it. This decoupled setup cleanly separates the modeling of the marginal distributions (via the autoencoders) from the bridging process, allowing each component to specialize in its subtask. However, bridging between arbitrary distributions in latent space presents two key challenges. First, because the bridging is performed entirely in the latent domain, fine-grained details, such as high-frequency visual information, may not be accurately preserved or reconstructed. Second, the autoencoders are trained independently and without explicit coupling, which can result in misaligned latent spaces and significant divergences between the encoded source and target distributions.

In Fig. 1, we qualitatively illustrate the limitations of the basic formulation through two analyses: super-resolution (left) and t-SNE embedding [60] plots of encoded ShapeNet representations [64] (right). On the left, the high-resolution reconstructions generated using the loss defined in Eq. (6) (labeled "Basic") exhibit noticeable artifacts and lack high-frequency details, such as the yellow hair ornament and fine facial features. On the right, the t-SNE plots show latent representations colored by semantic category. Circles denote encodings of multi-view images, while triangles represent encodings of 3D shapes. The center plot corresponds to the basic method, while the rightmost plot shows results from our approach. Although intra-modality clustering is relatively coherent in both cases, the basic method fails to align semantically similar samples across modalities, revealing a significant gap in cross-modality consistency within the latent space. To address these issues, we introduce a *predictive loss* that constrains the entire encode–bridge–decode pipeline, encouraging accurate domain translation. In addition, we incorporate a *contrastive loss* to attract semantically similar examples and repel dissimilar ones, as further detailed below.

**A predictive loss.** To improve the fidelity of the learned diffusion bridge, we introduce a predictive loss that enforces alignment between the input data and its reconstruction after being mapped through the bridge. Specifically, we encode the signal, apply the bridge transformation, decode the result, and compare it to the original input in pixel space. Formally,

$$\mathcal{L}_{\text{pred}} = d\left(D_x \circ B \circ E_y(y), x\right) \ , \tag{7}$$

where $B$ denotes the bridge model. This loss encourages the model to preserve semantic content across the full encode–bridge–decode pipeline, serving as a powerful regularizer for the bridge mapping. Importantly, as we empirically showcase in Sec. 5, replacing the two separate autoencoding losses previously used with this single predictive term for the source and target domains improves

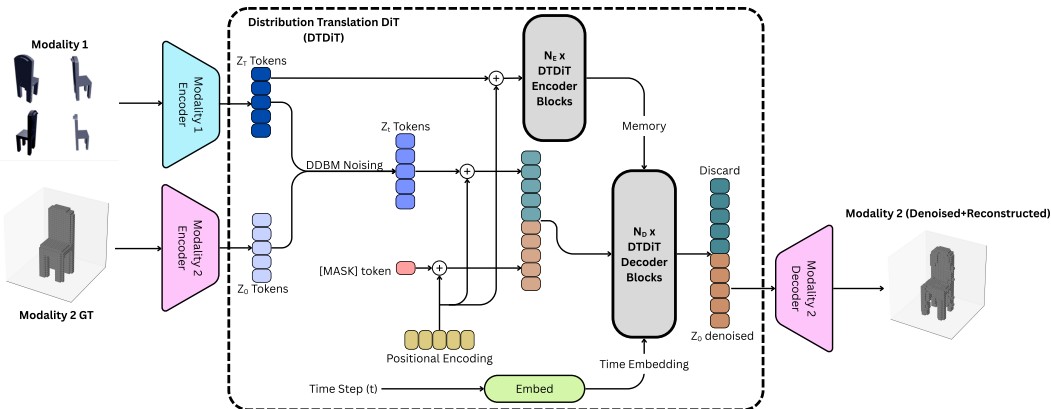

Figure 2: **End-to-end training pipeline for our framework**. During inference, $z_t$ is replaced with $z_T$ and Heun sampling is applied autoregressively to obtain $z_0$. For more details, see Fig. 5.

generative performance. As a result, we reduce computational overhead, shorten training time, provide one-way supervision, and improve the overall robustness and stability of the model.

**A contrastive loss.** Misaligned latent representations across independent distributions are typically addressed using alignment objectives. Common approaches include *cosine similarity* [48] and *contrastive estimation* [43]. Cosine similarity encourages similar samples to align by minimizing the angle between their embeddings, while contrastive estimation explicitly pulls together positive pairs and pushes apart negative pairs. In our setting, where we bridge source and target distributions, contrastive learning offers a natural fit. Leveraging the paired structure in the data, we treat $(z_0, z_T)$ as a positive pair and all other examples in the batch as negative examples. This setup allows us to effectively encourage alignment between related samples while preserving separation between unrelated ones. Formally, the loss is defined as

$$\mathcal{L}_{\text{infoNCE}} = \log \frac{\phi(z_0, z_T)}{\phi(z_0, z_T) + \sum_{j=1}^{M} \phi(z_0, z_T^j)} \ , \tag{8}$$

where $z_T^j$ are negative samples from the batch, and the function $\phi(u, v) = \exp(u^T v / \tau |u| |v|)$ measures the similarity between examples $u$ and $v$, with $\tau = 0.5$ being a temperature parameter [8, 41].

**Overall objective.** We incorporate the bridge, predictive and contrastive losses into a single unified objective forming our approach to training a bridge between arbitrary distributions

$$\mathcal{L} = \mathcal{L}_{\text{bridge}} + \mathcal{L}_{\text{pred}} + \mathcal{L}_{\text{infoNCE}} \ . \tag{9}$$

## 4.5 Multimodal Translation Architecture

In what follows, we motivate our neural architectural design by considering existing approaches and suggesting specific modifications to our bridging task. U-Nets and attention-based U-Nets [52, 19] are widely used in diffusion models. Recently, Transformer-based architectures such as DiT [45] have demonstrated superior performance using a decoder-only design. However, in many previous works such as [34, 39, 47, 31], the authors showcase the superiority of encoder-decoder transformer architectures over decoder-only architectures in the context of natural language translation tasks. Inspired by the effectiveness of the original encoder-decoder transformer architecture [61], we design a denoiser that is better suited for the distribution bridging task (see Fig. 2 and App. A.2.2).

Given the inputs $x$ and $y$, we first use modality-specific encoders to generate latent vectors $z_0$ and $z_T$, respectively, which are token sequences of the same embedding dimension [11, 48]. The details of the encoders used in our experiments are provided in App. A.2.1. In our design, we apply a transformer-encoder [61] to the tokens $z_T$, which outputs an embedding we term as *memory*. Unlike the original UNet implementation of DDBM, which concatenates this information to the input of the noised latent, our model consumes this memory using the cross-attention layers inside the transformer-decoder. This allows for more expressive conditioning. The transformer-decoder processes an input of the $z_t$ tokens concatenated with a sequence of learnable output tokens that are

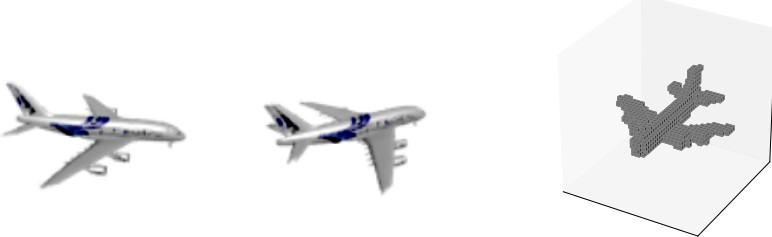

Figure 3: Multiview-to-3D-Shape translation example with *LDDBM*.

heavily inspired by context summarization tokens from previous transformer works such as ViT [11] and CLIP [48]. These learnable output tokens are obtained by adding the positional encoding to a learnable $[MASK]$ token. We want to highlight that the number of tokens in $z_t$ and $z_0$ are the same. To spatially align the output tokens with the $z_t$ tokens, we add the same positional encodings to both $z_t$ and $z_0$ to explicitly enforce spatial correspondence. The outputs of both the self- and cross-attentions and the feedforward networks in each transformer-decoder block are modulated using the timestep embedding of $t$ [45]. At the output of the transformer-decoder, we only use the output tokens in further loss computation. A modality decoder then processes the output estimate $\hat{z}_0$ to obtain $\hat{x}$. The *MASK* (or context-summarization) token is a special token used in many transformer architectures that attends to, and is attended by, all tokens, including under causal masking. It learns a single embedding summarizing the entire context window. This design parallels the **[CLS]** token in Vision Transformers (ViTs) for classification, where a prepended summary token serves as input to a downstream classifier. We empirically justify our design choices by comparing the effectiveness of our method to previous architectures in Tab. 4.

At inference time, we compute the memory embedding once for the whole Heun sampling process and use it for all subsequent denoising steps. We also replace $z_t$ with $z_T$ tokens for the first denoising step and then sample autoregressively, which is standard protocol to sample from a diffusion process.

**Training Procedure.** We experiment with several training regimes for our framework: (1) a two-step procedure, where we first train the autoencoders and then the diffusion bridge; (2) a fully end-to-end optimization; and (3) an iterative scheme that alternates between optimizing reconstruction and bridge alignment. The motivation for exploring the iterative regime stems from the inherent conflict between the bridge and the encoder networks. The bridge assumes that both marginal distributions remain fixed during training; however, since the encoders are trainable, they continuously reshape the structure of the latent distribution, effectively 'breaking' the previously learned bridge. This conflict of objectives resembles adversarial setups, which inspired us to incorporate practices from adversarial training [18, 44]. Among the three strategies, the iterative approach provides the best trade-off between training stability and final performance (see Sec. A.3.4).

## 5 Experiments

In this section, we present comprehensive experiments that evaluate our general-purpose framework for modality translation across a diverse set of tasks and domains. We benchmark our approach on four tasks and perform extensive ablation studies to assess the contribution of architectural components, loss functions, and training strategies. Due to space constraints, we include in App. A.3.1 the **Multiview-to-Scene-Occupancy Generation** task, in App. A.3.2 the **DIV2K Super-Resolution** task, and present a **complexity analysis** in App. A.3.3. Additional details regarding our experimental setup and implementation can be found in App. A.1 and App. A.2.

### 5.1 Multiview-to-3D-Shape Generation

We begin with the task of generating 3D shapes from 2D multi-view images. In this task, a general multimodal model is given multiple images of the same object from random viewpoints and their

Table 1: Quantitative comparison on the Multiview-to-3D-Shape Generation task.

|  | Pix2Vox-A | EDM | 3D-EDM | DiT | SiT | LDDBM |
|---|---|---|---|---|---|---|
| 1-NNA ↓ | - | .532 ± .013 | .575 ± .009 | .548 ± .004 | .563 ± .007 | **.508 ± .005** |
| IoU ↑ | 0.697 | .631 ± .006 | .602 ± .003 | .613 ± .011 | .604 ± .003 | **.664 ± .002** |

corresponding 3D shape during training. The goal of the model in inference time is to generate new 3D samples given multi-view samples that follow the distribution of 3D shapes.

**Evaluation Setup.** We evaluate our proposed approach using the ShapeNet dataset [64] following the protocol established in [55]. In this setup, each object is rendered into four 2D views captured from uniformly spaced azimuth angles around the object. These multi-view images form the input distribution $p(y)$ for our multimodal generative framework, and the 3D occupancy will serve as the output distribution $p(x)$. For evaluation, we adopt generative and reconstruction-based metrics to assess the quality of the models comprehensively. Note that our focus is on the generative performance of the models. For generative evaluation, we utilize the 1-NNA as a robust metric for assessing the similarity between the distributions of generated and real samples [74, 66]. To evaluate reconstruction quality, we employ IoU. The 1-NNA assesses how well the model captures the generative distribution (diversity and realism), while IoU quantifies fidelity to a specific target shape in reconstruction tasks. We compare our proposed method with several strong baselines capable of conducting general MT. Specifically, we include comparisons with the EDM and EDM-3D models [27], representing leading diffusion-based approaches using 2D and 3D U-Net architectures [52]. Additionally, we evaluate against recent Transformer-based diffusion models, including the Diffusion Transformer (DiT) [45] and the Scalable Interpolant Transformer (SiT) [35], both of which leverage transformer backbones to achieve strong performance on high-resolution generative tasks. For a fair comparison, we implement the same encoders and decoders as ours. Finally, while direct comparisons to task-specific SOTA may not be fully fair, as our framework is general-purpose, whereas those methods embed strong domain-specific inductive biases, we added a representative task-specialized SOTA, Pix2Vox-A [65] to the Multiview-to-3D Shape Generation table (highlighted in gray) to illustrate trade-offs.

**Results.** In Tab. 1, we present the quantitative results for the task of 3D shape generation from multi-view images, using both generative and reconstruction-based metrics. LDDBM achieves the lowest 1-NNA score (.**508**), significantly outperforming all baselines, indicating that our model generates 3D shapes more distributionally similar to the real data. Additionally, our model achieves the highest IoU score (.**664**), demonstrating superior fidelity in capturing fine-grained 3D geometry compared to all competing methods. Overall, LDDBM consistently outperforms strong baselines across both evaluation axes, confirming its strong generative capabilities and its ability to generate 3D content from multi-view images faithfully. Further, we show qualitative results in Fig. 3 and in App. A.3, and comparison in A.3.8, showcasing the input multi-view images alongside its corresponding 3D shape.

## 5.2 Zero-Shot Low-to-High Resolution Generation

To further explore our model's ability to generalize across different modality generations, we evaluate the zero-shot super-resolution capabilities of our model since images lie in different modality sizes. This task aims to generate high-resolution images from low-resolution inputs without task-specific fine-tuning. In this setup, the general multimodal generative model is trained on paired low-resolution and high-resolution images. During inference, the model receives only a low-resolution image and must produce a realistic high-resolution output that conforms to the distribution of real images.

**Evaluation Setup.** We follow the experimental protocol of [38]. The model is trained on the Flickr-Faces-HQ (FFHQ) dataset [28] and evaluated on samples from the CelebA-HQ dataset [26]. The task is defined from a resolution of $16 \times 16$ to $128 \times 128$. To evaluate the perceptual and pixel-level quality of generated images, we follow the experimental protocol of [38], which uses the metrics: Peak Signal-to-Noise Ratio (PSNR), Structural Similarity Index (SSIM), and Learned Perceptual Image Patch Similarity (LPIPS). These metrics provide a comprehensive evaluation: PSNR and SSIM capture low-level similarity, while LPIPS captures high-level perceptual realism, which is particularly relevant for generative tasks. We compare our method with a similar baseline as the previous section and a task-specific state-of-the-art diffusion-based baseline DiWa [38].

**Results.** Fig. 4 (bottom) shows the quantitative results. Our model yields a PSNR of **25**.**6**, surpassing other methods and indicating high reconstruction fidelity. Similarly, our model yields the highest SSIM of **0**.**68**, demonstrating its superior ability to preserve structural information. Importantly, our model achieves the lowest LPIPS score of **0**.**32**, highlighting its strength in generating perceptually realistic images, as judged by deep feature similarity. The consistent improvement across pixel-wise, structural, and perceptual metrics demonstrates the robustness and versatility of our approach in zero-shot generative tasks. Additionally, we provide qualitative results in Fig. 4 (top) and App. A.3 , and comparisons in App. A.3.8, showcasing high-quality samples produced by our model.

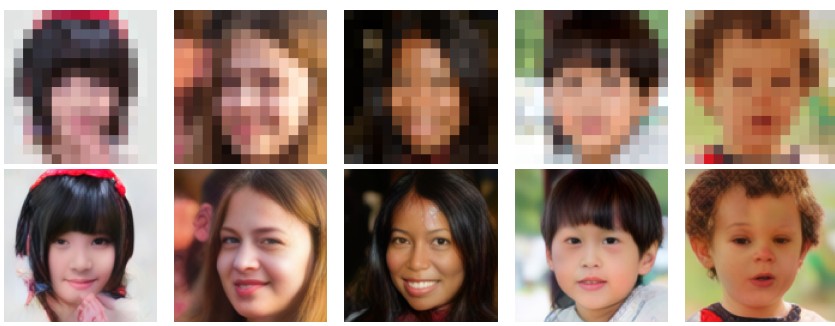

| | EDM | DIWA | DIT | SIT | LDDBM |
|---|---|---|---|---|---|
| PSNR ↑ | $23.1 \pm 0.7$ | 23.3 | $22.2 \pm 1.1$ | $21.5 \pm 0.4$ | $\mathbf{25.6 \pm 0.4}$ |
| SSIM ↑ | $0.58 \pm .05$ | 0.65 | $0.52 \pm .07$ | $0.57 \pm .02$ | $\mathbf{0.68 \pm .03}$ |
| LPIPS ↓ | $0.41 \pm .02$ | 0.39 | $0.49 \pm .01$ | $0.51 \pm .03$ | $\mathbf{0.32 \pm .01}$ |

Figure 4: Top: Example low-to-high resolution outputs by LDDBM. Bottom: Zero-shot quantitative comparison measures. (Standard deviations could not be extracted for DiWa.)

## 5.3 Voice-to-Face and Face-to-Voice Translation

To further diversify our benchmarking and strengthen the robustness of our results, we introduce two additional multimodal translation tasks: *Image-to-Audio* and *Audio-to-Image*, following the benchmark proposed by [40]. In these tasks, a multimodal translation model receives one modality (either face or voice) as input and generates a meaningful representation in the other modality, which is subsequently evaluated through downstream classification accuracy. We compare our method against two baselines: (i) the SiT variant, which achieved the overall second-best performance on other multimodal benchmarks, and (ii) the specialized state-of-the-art method from [40], specifically tailored for cross-modal biometric matching between faces and voices. Implementation and evaluation details follow the same protocol as in [40], and full experimental details, including encoder and decoder architectures, are provided in the appendix.

Tab. 2 presents the results, where higher values indicate better accuracy. LDDBM consistently outperforms the general-purpose SiT model, demonstrating stronger multimodal translation capability. Although a performance gap remains compared to the task-specific approach of [40], this is expected given its domain specialization. Overall, the results highlight the generality of our framework, its superiority among modality-agnostic baselines, and its adaptability to diverse modalities.

Table 2: Cross-modal biometric translation accuracy on face–voice matching tasks. Higher values indicate better performance.

| Method | Face → Voice ↑ | Voice → Face ↑ |
|---|---|---|
| *LDDBM* | **71.2** | **75.1** |
| SiT | 65.7 | 68.3 |
| [40] | 79.5 | 81.0 |

Table 3: **Edges → Bags (image-to-image).** FID (lower is better) and inference time (seconds, lower is better). Measured with batch size 256.

| Method / Metric | FID ↓ | Time (s) ↓ |
|---|---|---|
| *LDDBM* | 4.17 | 7.8 |
| DDBM | 2.93 | 16.9 |

## 5.4 Image-to-Image Translation

While our benchmark was designed to target translation between *different-dimensional* modalities, it is also important to assess tasks with *aligned* modality dimensions to further evaluate robustness and generalizability. To this end, we consider the *edges → bags* task from the DDBM benchmark [75], which directly targets image-to-image translation. We evaluate two aspects: sample quality (FID ↓) and inference performance (wall-clock time in seconds ↓). Results are reported in Table 3. Overall, we observe a clear trade-off: our method is competitive in terms of quality while providing more than $2\times$ faster inference than DDBM.

Table 4: **Step-by-step ablation of our Transformer architecture**. Each row introduces a new component, illustrating cumulative improvements and the contribution of each design choice. The full results, with standard deviations, are reported in App. A.3.

| Component | ShapeNet | | nuScene | | CelebA-HQ | | |
|---|---|---|---|---|---|---|---|
| | IoU ↑ | 1-NNA ↓ | IoU ↑ | 1-NNA ↓ | PSNR ↑ | SSIM ↑ | LPIPS ↓ |
| (1) U-Net | .635 | .518 | .216 | .818 | 23.2 | 0.57 | 0.42 |
| (2) DiT | .613 | .548 | .208 | .825 | 22.2 | 0.52 | 0.49 |
| (3) + Encoder-Decoder | .651 | .518 | .217 | .821 | 23.4 | 0.53 | 0.38 |
| (4) + Spatial Embedding | .658 | .522 | .224 | .812 | 22.9 | 0.56 | 0.41 |
| (5) + $[MASK]$ (Ours) | **.664** | **.508** | **.233** | **.807** | **25.6** | **0.68** | **0.32** |

Table 5: Loss ablation: Comparison of reconstruction and contrastive losses.

| | | $\mathcal{L}_{\text{REC}}$ | $\mathcal{L}_{\text{PRED}}$ | $\mathcal{L}_{\text{REC}} + \mathcal{L}_{\text{INFONCE}}$ | $\mathcal{L}_{\text{PRED}} + \mathcal{L}_{\text{INFONCE}}$ |
|---|---|---|---|---|---|
| SHAPENET | 1-NNA ↓ | .625 ± .003 | .522 ± .002 | .578 ± .004 | **.508 ± .005** |
| | IoU ↑ | .609 ± .007 | .643 ± .005 | .627 ± .007 | **.664 ± .002** |
| CELEBA-HQ | PSNR ↑ | 20.5 ± 0.4 | 23.7 ± 0.3 | 21.4 ± 0.1 | **25.6 ± 0.4** |
| | SSIM ↑ | 0.49 ± .03 | 0.64 ± .02 | 0.51 ± .05 | **0.68 ± .03** |
| | LPIPS ↓ | 0.62 ± .04 | 0.41 ± .02 | 0.63 ± .03 | **0.32 ± .01** |

## 5.5 Ablation Studies

**Architectural ablation.** We conduct a thorough ablation study of the architectural components in our general-purpose Transformer. Additional ablations of different training procedures can be found in App. A.3.4. As shown in Table 4, we begin with a U-Net backbone (1), which leverages spatial inductive biases but lacks generality across modalities. Switching to DiT (2) removes these biases, revealing a performance gap. To improve conditioning, we adopt an encoder-decoder design (DiT is decoder only architecture) (3) inspired by sequence-to-sequence translation models. We introduce constant spatial embeddings (4) to align tokens between input and target modalities. Finally, we incorporate a learnable $[MASK]$ token (5) to enhance expressiveness and achieve state-of-the-art performance. We observe minor performance improvements in specific datasets on certain metrics, where spatial embedding is added (4), suggesting that regular positional embedding (3) and spatial embedding (4) can be used interchangeably based on preference or prior biases.

**Losses ablation.** We additionally assess the impact of various loss configurations. We compare:

1. **Reconstruction loss:** Standard reconstruction $\mathcal{L}_{\text{rec}} = \mathcal{L}_{\text{AE}_x} + \mathcal{L}_{\text{AE}_y}$ vs. consistent reconstruction through the multi-modal bridge $\mathcal{L}_{\text{pred}}$.

2. **Contrastive loss:** With or without additional contrastive estimation $\mathcal{L}_{\text{infoNCE}}$.

The results summarized in Table 5 indicate that the complete configuration combining both $\mathcal{L}_{\text{pred}}$ and $\mathcal{L}_{\text{infoNCE}}$ achieves the highest performance. In addition, Fig. 10 qualitatively illustrates the differences between the $\mathcal{L}_{rec} + \mathcal{L}_{infoNCE}$ and $\mathcal{L}_{\textbf{pred}} + \mathcal{L}_{\textbf{infoNCE}}$ models. It is evident that the outputs of the $\mathcal{L}_{rec} + \mathcal{L}_{infoNCE}$ model suffer from facial artifacts, distorted backgrounds, and irregular distributions, while the $\mathcal{L}_{\textbf{pred}} + \mathcal{L}_{\textbf{infoNCE}}$ model produces images that are perceptually cleaner and more reliable.

## 6 Conclusion

In this work, we introduced a general framework for modality translation based on latent-variable extensions of Denoising Diffusion Bridge Models. By operating in a shared latent space and leveraging contrastive alignment and predictive objectives, our method enables flexible and accurate translation between arbitrary modalities without relying on restrictive assumptions such as aligned dimensions or modality-specific architectures. Through comprehensive experiments across diverse tasks—including multi-view 3D shape generation, image super-resolution, and multi-view scene synthesis—we demonstrated strong performance and robust generalization, establishing a new state-of-the-art for general modality translation. Looking forward, extending our framework to handle unpaired modality translation and scaling it to sequential or high-dimensional data such as video and volumetric representations offers promising directions for further expanding its applicability.

## Acknowledgments

This research was partially supported by the Lynn and William Frankel Center of the Computer Science Department, Ben-Gurion University of the Negev, an ISF grant 668/21, an ISF equipment grant, and by the Israeli Council for Higher Education (CHE) via the Data Science Research Center, Ben-Gurion University of the Negev, Israel.

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

# A    Appendix / supplemental material

## A.1    Additional Experimental Details

### A.1.1    Datasets

**Multi-view to 3D generation task - ShapeNet**    [64] is a comprehensive dataset comprising tens of thousands of unique 3D models across 55 common object categories. In alignment with the protocol established in [55], we utilize a subset of this dataset, focusing on 13 specific categories, encompassing around 43,783 models. Each 3D model undergoes preprocessing with Binvox at a resolution of $32 \times 32 \times 32$. Subsequently, 4 images are rendered per model from random viewpoints, each with a resolution of $224 \times 224$ pixels. We split the dataset by randomly assigning 70% of the objects to the training set, 10% to the validation set, and the remaining 20% to the test set, ensuring a balanced distribution across all categories.

**Zero-Shot Low-to-High Resolution Generation task - FFHQ and CelebA-HQ:**    The FFHQ dataset comprises 70,000 high-quality images of human faces at a resolution of 1024×1024 pixels, for our task, the images where resized to $16 \times 16, 128 \times 128$ pair with Bilinear interpolation. These images exhibit considerable variation in age, ethnicity, and background, and include a diverse range of accessories such as eyeglasses, sunglasses, and hats. This dataset was used to train our model.

The CelebA-HQ dataset is a high-quality version of the original CelebA dataset, consisting of 30,000 images at a resolution of 1024×1024 pixels, for our task, the images where resized to $16 \times 16, 128 \times 128$ pair with Bilinear interpolation.. It was created by selecting and processing images from CelebA to enhance their quality and resolution. This dataset was used to evaluate the model

**Multiview-to-Scene-Occupancy Generation - nuScenes-Occupancy**    The dataset serves as a 3D occupancy prediction benchmark derived from the nuScenes autonomous driving dataset. It contains approximately 25,000 pairs of multi-view camera images and corresponding 3D occupancy representations of the ego vehicle's surroundings. The occupancy information is inferred from LiDAR data and discretized into a $512 \times 512 \times 32$ voxel grid. Each voxel grid is paired with a set of high-resolution multi-view images captured from cameras mounted on the vehicle, covering a full $360°$ field of view. To reduce computational demands, we downsample the original high-resolution images to $256 \times 256$, as processing the full-resolution inputs would require a minimum of $80$ GB of memory per sample, which is impractical given resources limitations.

### A.1.2    Metrics

**Chamfer Distance**    is used to calculate distance metric for the 1-NNA metric. Therefore, we do not directly apply this as a metric. This distance is used to compare the similarity between two point clouds, typically between a generated 3D shape and a reference. It is defined as:

$$\text{CD}(\mathcal{P}, \mathcal{Q}) = \frac{1}{|\mathcal{P}|} \sum_{p \in \mathcal{P}} \min_{q \in \mathcal{Q}} \|p - q\|_2^2 + \frac{1}{|\mathcal{Q}|} \sum_{q \in \mathcal{Q}} \min_{p \in \mathcal{P}} \|q - p\|_2^2$$

where $\mathcal{P}$ and $\mathcal{Q}$ are two point clouds. Lower Chamfer Distance indicates higher geometric similarity.

**1-Nearest Neighbor Accuracy (1-NNA)**    is a robust metric for assessing the similarity between the distributions of generated and real samples [66, 74]. The 1-NNA metric is grounded in a two-sample statistical test and has been proven to provide a more reliable measure of distributional similarity than commonly used heuristics which may be sensitive to mode collapse or fail to capture full-shape distributions.

Formally, let $\mathcal{S}_{\text{real}}$ and $\mathcal{S}_{\text{gen}}$ denote the sets of real and generated samples, respectively. A 1-NN classifier is trained over the combined dataset $\mathcal{S} = \mathcal{S}_{\text{real}} \cup \mathcal{S}_{\text{gen}}$, with binary labels indicating whether each sample is real or generated. For each sample $x \in \mathcal{S}$, its nearest neighbor $\text{NN}(x)$ is found among the rest of the set $\mathcal{S} \setminus \{x\}$, and a leave-one-out classification is performed. The 1-NNA score is defined as:

$$1\text{-NNA} = \frac{1}{|\mathcal{S}|} \sum_{x \in \mathcal{S}} \mathbb{I}\left[\text{label}(x) = \text{label}(\text{NN}(x))\right],$$

where $\mathbb{I}[\cdot]$ is the indicator function. Distances are computed using Chamfer Distance to reflect geometric dissimilarity between point clouds.

An ideal generative model produces samples indistinguishable from real data, which would result in a 1-NNA score of 50%. Scores significantly above 50% indicate that the classifier can distinguish between real and generated samples, thus revealing discrepancies between the distributions. Therefore, *lower is better* for this metric, and values closer to 50% reflect stronger generative performance.

Since our data is voxelized, we first convert each object into a point cloud representation in order to apply the metric. We then use the standard evaluation protocol interface to compute the final results.

**Intersection over Union (IoU)** aims to evaluate reconstruction quality, and its a standard metric in 3D shape prediction. For voxelized 3D shapes, IoU is computed between the predicted occupancy grid $\hat{V}$ and the ground truth grid $V$ as:

$$\text{IoU} = \frac{|\hat{V} \cap V|}{|\hat{V} \cup V|}.$$

A higher IoU indicates better overlap with the true shape, reflecting accurate reconstructions. This metric emphasizes pixel-wise consistency and is most relevant when the goal is to reproduce specific ground-truth instances.

**Peak Signal-to-Noise Ratio (PSNR):** This metric measures the pixel-wise similarity between the generated image $\hat{x}$ and the ground truth image $x$, computed as:

$$\text{PSNR}(x, \hat{x}) = 10 \cdot \log_{10} \left( \frac{MAX^2}{\frac{1}{N} \sum_{i=1}^{N} (x_i - \hat{x}_i)^2} \right)$$

where $MAX$ is the maximum pixel value (e.g., 255) and $N$ is the number of pixels. Higher PSNR indicates lower mean squared error and better reconstruction fidelity.

**Structural Similarity Index (SSIM):** SSIM evaluates the structural similarity between two images, capturing luminance, contrast, and texture similarity. It is defined as:

$$\text{SSIM}(x, \hat{x}) = \frac{(2\mu_x \mu_{\hat{x}} + C_1)(2\sigma_{x\hat{x}} + C_2)}{(\mu_x^2 + \mu_{\hat{x}}^2 + C_1)(\sigma_x^2 + \sigma_{\hat{x}}^2 + C_2)}$$

where $\mu$, $\sigma$ are means and variances, and $C_1$, $C_2$ are constants for numerical stability. Higher SSIM values (closer to 1) indicate better structural fidelity.

**Learned Perceptual Image Patch Similarity (LPIPS):** LPIPS [71] measures perceptual similarity using deep neural network features. It compares deep activations between $x$ and $\hat{x}$:

$$\text{LPIPS}(x, \hat{x}) = \sum_l \frac{1}{H_l W_l} \sum_{h,w} \|w_l \odot (f_l^x - f_l^{\hat{x}})\|_2^2$$

where $f_l$ are the features from layer $l$, $w_l$ is a learned weight vector, and $H_l, W_l$ are spatial dimensions. Lower LPIPS implies higher perceptual similarity.

## A.2 Implementation Details

### A.2.1 Modality Encoder and Decoder Architectures

The encoders and decoders are similar between different methods, providing a more fair comparison setup between different general multi-modal generative frameworks. Therein, we describe the specific of each encoder or decoder that we use in the different tasks.

**Multiview-to-3D-Shape Generation** In this setup, except for the 3D-EDM method, all frameworks use the same encoder decoder scheme.

1. **3D Convolutional Encoder -** To encode 3D object inputs, we adopt a 3D convolutional encoder. The encoder takes as input a tensor of shape $(B, C, D, H, W)$, where $B$ is the batch size, $C$ the number of channels, and $H \times W \times D$ the 3D-voxel resolution of the object. The network comprises a sequence of three 3D convolutional layers with kernel size 4, stride 2, and padding 1, progressively reducing the spatial dimensions. Specifically, the first layer maps the input to 64 channels, followed by a second and third layer expanding the channels to 128 and 256, respectively. Each convolution is followed by a SiLU (Sigmoid Linear Unit) activation function and optional dropout for regularization. The final output is flattened or reshaped into a latent tensor, depends on the model requirements. This latent representation serves as the encoded feature map for downstream generative modeling tasks.

2. **3D Convolutional Decoder -** This module mirrors the architecture of the 3D encoder, but uses transposed convolutional layers to progressively upsample the latent representation back to the original volumetric shape.

3. **Multi-View 2D Convolutional Encoder -** To process multi-view image inputs, we utilize a 2D convolutional encoder that independently encodes each view using a shared backbone. The encoder receives input tensors of shape $(B, V, C, H, W)$, where $B$ is the batch size, $V$ is the number of views, $C$ the number of image channels (e.g., RGB), and $H \times W$ the resolution of each view. Each view is flattened across the batch and view dimensions and passed through a shared 2D convolutional stack consisting of six layers. Each layer applies a convolution with kernel size 4, stride 2, and appropriate padding, followed by a SiLU (Sigmoid Linear Unit) activation and optional dropout for regularization. The convolutional stack progressively increases the channel dimension from 32 to 256 while reducing the spatial resolution, effectively extracting high-level features from each view.

   Following the convolutional encoding, the features from all views are concatenated and passed through a fully connected projection layer to form a unified latent representation. Specifically, the concatenated features are mapped to a latent space of configured size using a linear layer.

4. **Identity Encoder/Decoder for 3D Data -** This is a special case used in the 3D-EDM model, where the diffusion process operates directly in the input space $x$. As a result, no encoding or decoding modules are required, and the model functions similarly to standard diffusion models without latent-space adaptation.

**Zero-Shot Low-to-High Resolution Generation** In this setup, for all methods, including ours, except for [38], whose results were taken from the benchmark, we used the same encoder-decoder architecture. Specifically, we employed the pre-trained autoencoder from [50] with the configuration $f = 32$, KL regularization, and latent dimension $d = 64$. All models were initialized with the pre-trained weights before training.

To handle varying image resolutions (as the autoencoder was originally trained on $256 \times 256$ images), we prepend each model with a stack of 2D convolutional upsampling layers to resize inputs to $256 \times 256$, and append a symmetric stack of downsampling layers to revert to the original resolution. The latent representations used for generative modeling of distribution translation.

**Multiview-to-Scene-Occupancy Generation** In this setup, we adopt a *3D convolutional encoder-decoder* architecture for encoding and decoding the occupancy scene, following a similar design to that used in multi-view to 3D shape generation. For encoding the multi-view images, we utilize the encoder from the pre-trained autoencoder proposed in [50].

### A.2.2 Our Transformer Architecture

Fig. 5 shows the structure and order of operations performed by the Distribution Translation DiT (DTDiT). The DTDiT Encoder block (Fig. 5b) consists of self-attention and feed-forward operations, each preceded by a layernorm, exactly like the transformer-encoder shown in [61]. In the DTDiT Decoder block (Fig. 5a), we extend the original DiT Block [45], by adding a time-embedding modulated cross-attention operation, in between the original self-attention and feed-forward operations. Consequently, the MLP learns 9 vectors ($\alpha_i, \beta_i, \gamma_i \quad \forall i \in \{1, 2, 3\}$) instead of the original 6 ($\alpha_i, \beta_i, \gamma_i \quad \forall i \in \{1, 2\}$).

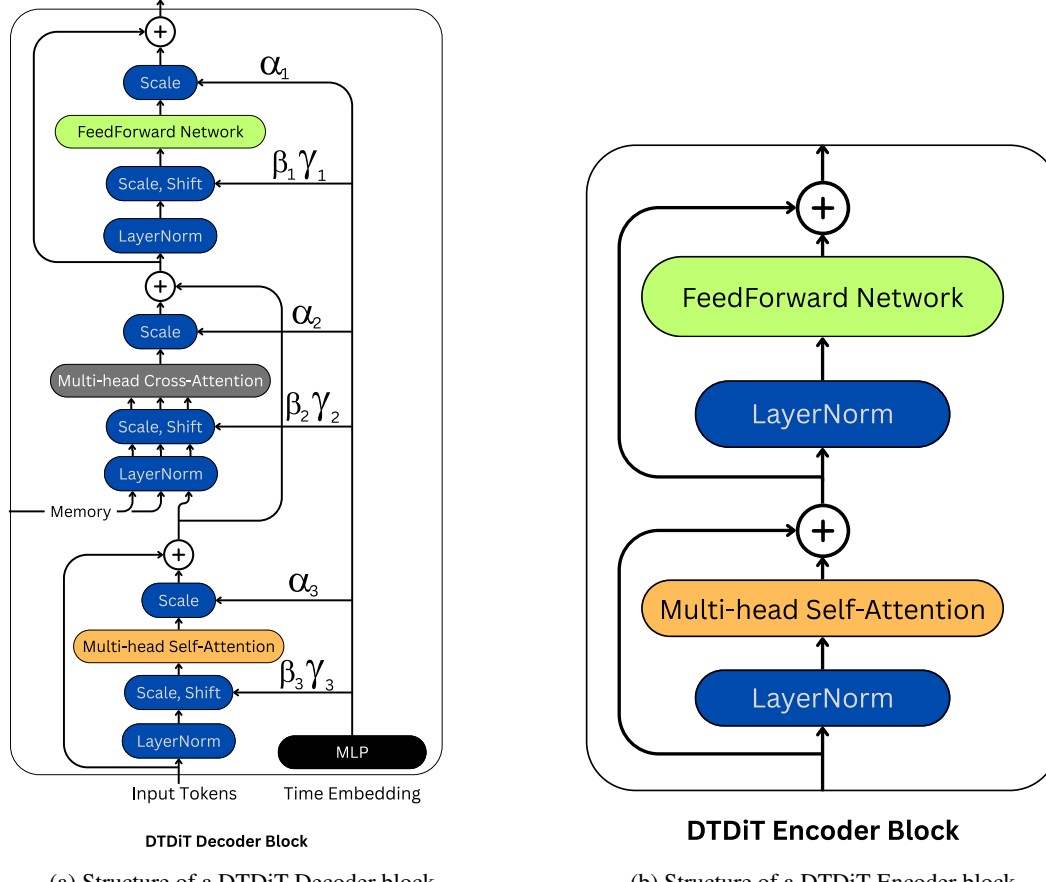

(a) Structure of a DTDiT Decoder block       (b) Structure of a DTDiT Encoder block

Figure 5: Detailed network architecture of the Distribution Translation DiT (DTDiT)

### A.2.3 Baselines Implementation

For all baselines, we employed a similar encoding and decoding scheme to ensure a fair comparison for general modality translation (MT) evaluation. To enable general MT across all baselines [35, 45, 27], we used the encoding of $y$, denoted $z_T$, as the conditional input to the diffusion model, and the encoding of $x$, denoted $z_0$, as the target signal to be generated. The diffusion process was then trained to generate $z_0$ conditioned on $z_T$ using the standard diffusion loss, along with an additional reconstruction loss to map the predicted latent $z$ back to its original domain. We evaluated both predictive and reconstruction-based losses for each method, reporting the better-performing one.

During inference, we used the original sampler of each baseline, but standardized the number of sampling steps to 40 for consistency. For each method, we retained the original architectural hyperparameters and only adjusted the latent space dimensionality to match our setup - except for EDM-3D (on the ShapeNet dataset), which operates directly in the same dimension as $x$. All methods used the same latent dimensionality as our framework.

### A.2.4 Hyperparameters

We adopted the Variance Exploding (VE) configuration for sampling, following the hyperparameter settings provided in the official codebase of [76]. The hyperparameters we tuned are summarized in Table 6. Our search focused on variations in batch size, learning rate, and latent space dimensionality. All loss terms were used without weighting adjustments.

Training was performed on a single NVIDIA A100 GPU for 1,000,000 iterations, which corresponds to approximately 3–5 days of runtime depending on the dataset. We used the RAdam optimizer.

Table 6: Summary of Hyperparameters

|  | MULTI-VIEW-TO-3D-SHAPE | ZERO-SHOT-SUPER-RESOLUTION | MULTI-VIEW-TO-SCENE |
|---|---|---|---|
| BATCH SIZE | 256 | 32 | 64 |
| LEARNING RATE | 0.0001 | 0.0003 | 0.0001 |
| LATENT SPACE SIZE | 16384 | 24576 | 4096 |
| SAMPLING STEPS | 40 | 40 | 40 |
| RHO | 7 | 7 | 7 |
| SAMPLER | HEUN | HEUN | HEUN |
| CHURN STEP RATIO | 0 | 0 | 0 |
| GUIDANCE | 0.5 | 0.5 | 0.5 |
| IN CHANNELS | 256 | 64 | 384 |
| EMBED DIM | 256 | 256 | 256 |
| NUM HEADS | 8 | 8 | 8 |
| DEPTH ($N_D$, $N_E$) | 6 | 6 | 6 |

### A.3 Additional Experiments and Details

#### A.3.1 Multiview-to-Scene-Occupancy Generation

Table 7: 1-NNA and IoU metrics for multi-view to 3D occupancy on the nuScene dataset.

| | EDM | 3D-EDM | DiT | SiT | LDDBM |
|---|---|---|---|---|---|
| 1-NNA ↓ | $.822 \pm .009$ | $.832 \pm .012$ | $.825 \pm .007$ | $.854 \pm .005$ | $\mathbf{.807 \pm .008}$ |
| IoU ↑ | $.213 \pm .007$ | $.202 \pm .013$ | $.208 \pm .011$ | $.198 \pm .007$ | $\mathbf{.233 \pm .005}$ |

We conclude our evaluation by considering a more complex task: generating 3D occupancy scenes from multi-view images using the nuScenes-Occupancy framework [7, 62] for occupancy prediction. This subset of nuScene dataset offers multi-camera views along with voxelized 3D occupancy map. For computational feasibility, we downsample both input images and occupancy volumes. While this experiment is not designed to push state-of-the-art boundaries, it showcases the flexibility and applicability of our framework in more realistic autonomous driving scenarios. We adopt the metric introduced in the previous Sec. 5.1. For all these methods, and for a fair comparison, we implement the same encoders and decoders as ours for embedding $x, y$. We report further details in App. A.2.

**Results.** We report the results in Table 7. Our method achieves the best performance across both evaluation metrics for the task. Specifically, it obtains the lowest 1-Nearest Neighbor Accuracy (1-NNA) of $\mathbf{0.807}$, indicating that the distribution of generated 3D shapes is closest to that of the real data in comparison to other methods. This reflects the generative quality in terms of realism and diversity. Nevertheless, these results indicate that all models still have a significant gap toward producing realistic generations. In addition, our model achieves the highest Intersection over Union (IoU) score of $\mathbf{0.233}$, significantly outperforming all baselines. This demonstrates superior reconstruction accuracy in capturing the underlying 3D structure. Taken together, these results highlight again the robustness and effectiveness of our approach.

#### A.3.2 Generalization to Natural Images: DIV2K Super-Resolution

To assess the generalization ability of our framework beyond face-centric data, we incorporated the **DIV2K** dataset, a large and diverse collection of high-quality RGB natural images, into the Super-Resolution (SR) benchmark. We trained and evaluated our model following the same setup as in [38].

**Results.** Qualitatively, as showen in Fig. 9, our model produces sharp, artifact-free images across a wide variety of natural scenes, demonstrating strong generalization beyond facial domains. Quantitatively, our approach achieves a PSNR of **28.9**, outperforming the DiWa baseline [38], which attains a PSNR of 28.2 under identical settings. These findings reinforce the flexibility of our latent bridging framework and its ability to generalize to domains with diverse textures and structures.

#### A.3.3 Complexity Analysis

Table 8 presents a comparison of model complexity and inference efficiency for different approaches evaluated on the ShapeNet dataset. We report the number of trainable parameters (in total) as well as the sampling time (in seconds) for a single batch of size 256. Our model demonstrates a clear advantage in terms of both model size and runtime efficiency. Specifically, it requires only **288M parameters**, significantly fewer than both EDM (**498M**) and DiT (**685M**), representing a reduction of over 40% compared to EDM. This leaner architecture implies lower memory footprint and easier deployment. Moreover, our model achieves a sampling time of **12.46 seconds**, which is more than **7 times faster** than the DiT-based model (**79.36 seconds**) while maintaining strong generative quality. It is also comparable to the convolutional EDM model in speed, yet substantially more compact and versatile. These results underscore the scalability and efficiency of our framework for high-resolution 3D generation tasks.

**Sampling complexity analysis.** We evaluated sampling quality on SHAPENET while varying the number of diffusion steps from 40 down to 10 (Table 9). Increasing the step count beyond 40 did not

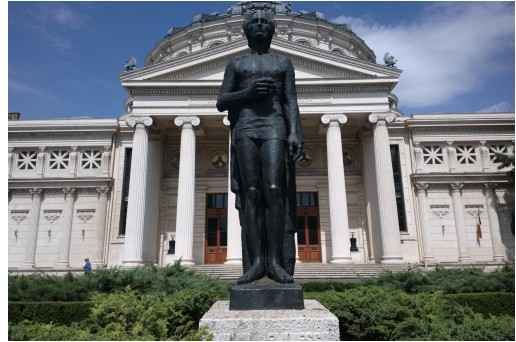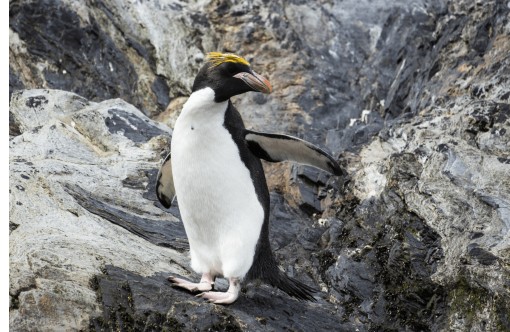

Figure 6: Original High Resolution

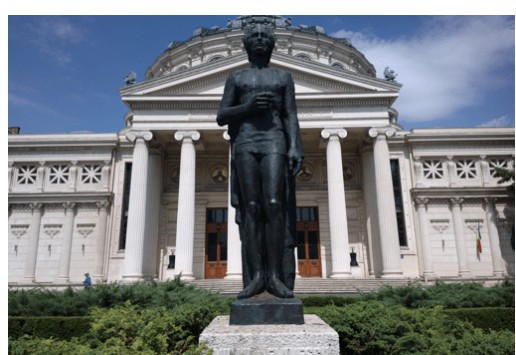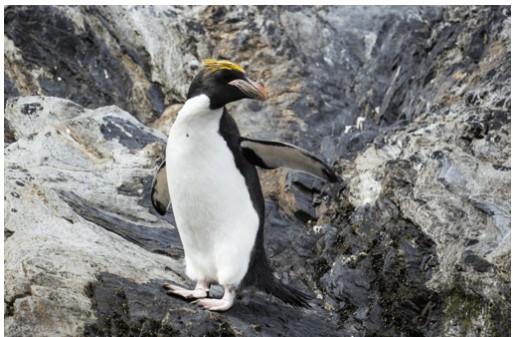

Figure 7: Low Resolution

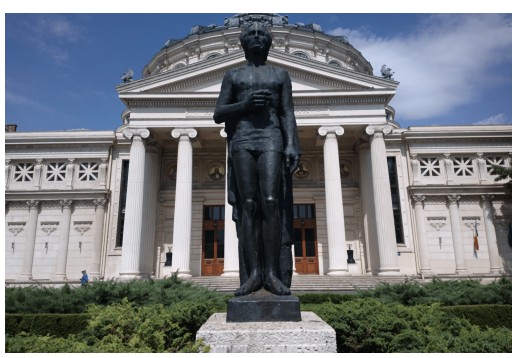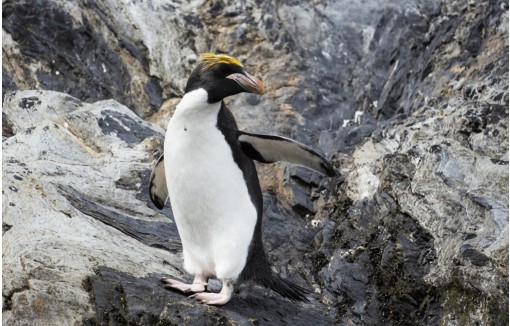

Figure 8: Our Super Resolution

Figure 9: **DIV2K super-resolution examples.**

Table 8: Complexity analysis on the ShapeNet dataset. Parameters are in millions. Sampling time is measured in seconds per batch of size 256.

|  | EDM | DiT | LDDBM |
|---|---|---|---|
| PARAMETRES ↓ | 498.6 | 685.5 | **288.7** |
| SAMPLING TIME ↓ | **11.17** | 79.36 | 12.46 |

yield measurable gains. Notably, reducing from 40 to 30 steps slightly *improves* quality, whereas using fewer than 30 steps leads to an apparent degradation, as expected.

| Model / steps | 40 | 30 | 20 | 10 |
|---|---|---|---|---|
| Our 1-NNA ↓ | $508 \pm .005$ | $504 \pm .006$ | $522 \pm .008$ | $631 \pm .018$ |
| IoU ↑ | $664 \pm .002$ | $663 \pm .004$ | $642 \pm .004$ | $533 \pm .014$ |

Table 9: **Sampling-step ablation on SHAPENET.** Performance as a function of the number of sampling steps. Values are mean $\pm$ std.

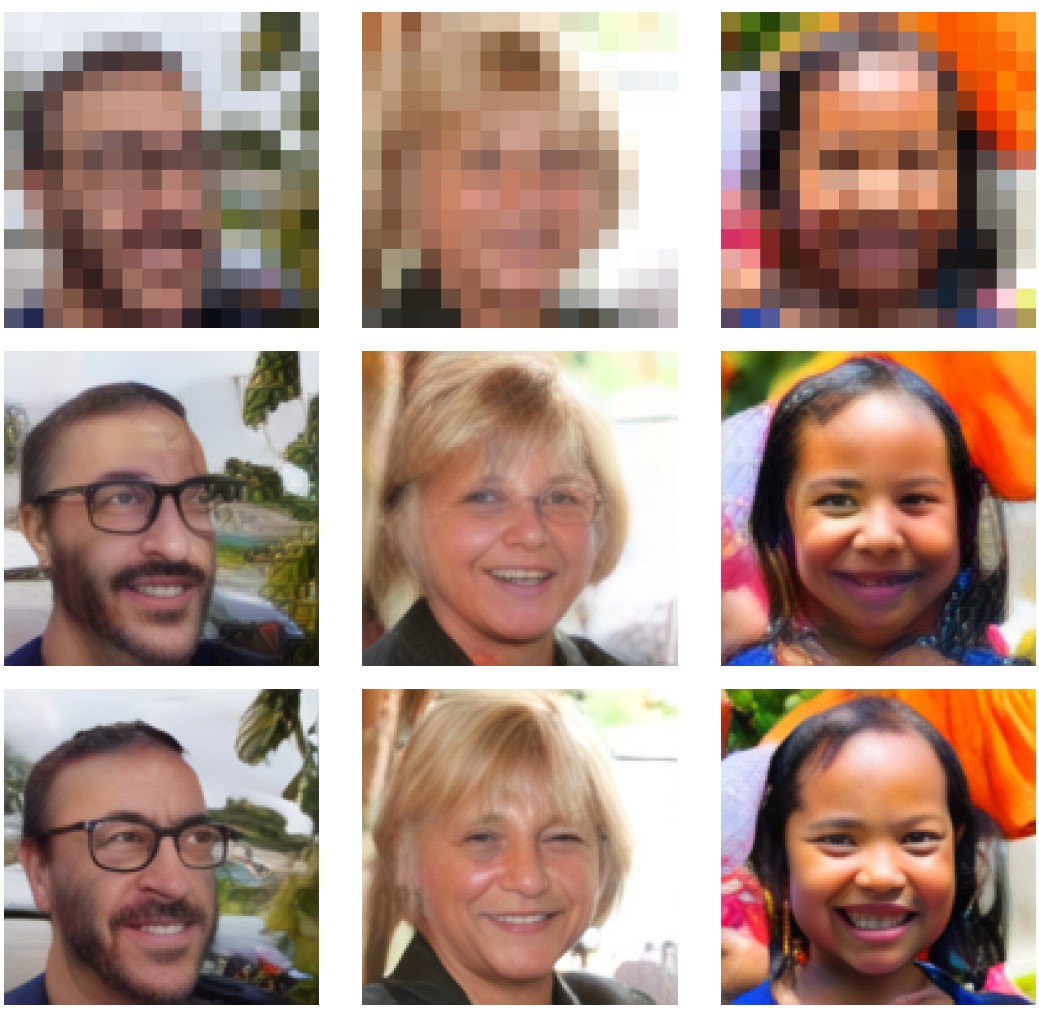

Figure 10: Evaluation of our model with different loss functions. The first row shows the low-resolution input, the second row displays the output using the regular reconstruction loss ($\mathcal{L}_{rec}$) + $\mathcal{L}_{infoNCE}$, and the last row presents the output using the end-to-end loss ($\mathcal{L}_{\mathbf{pred}} + \mathcal{L}_{\mathbf{infoNCE}}$).

### A.3.4 Training Strategies Ablation

We compare three training paradigms for multi-modal generation: *Two-Step* training, *End-to-End* training of the whole system, and our proposed *Iterative* procedure inspired by practices in adversarial training [18, 44]. As shown in Table 10, the Iterative strategy consistently yields the best results across datasets and metrics.

**Runtime setup and observations.** Unless stated otherwise, all runs use identical hardware and hyperparameters on a single NVIDIA A100 GPU. For both **End-to-End** and **Iterative** strategies,

Table 10: Comparison of training strategies: Two-Step, End-to-End, and Iterative.

| | | Two-Step | End-to-End | Iterative |
|---|---|---|---|---|
| ShapeNet | 1-NNA ↓ | .522 ±.006 | .517 ±.003 | **.508 ± .005** |
| | IoU ↑ | .637 ±.003 | .642 ±.005 | **.664 ± .002** |
| CelebA-HQ | PSNR ↑ | 23.3 ±0.6 | 23.4 ±0.3 | **25.6 ± 0.4** |
| | SSIM ↑ | 0.58 ±.07 | 0.57 ±.05 | **0.68 ± .03** |
| | LPIPS ↓ | 0.40 ±.02 | 0.39 ±.01 | **0.32 ± .01** |

we train for a total of 1,000,000 iterations. The **Two-Step** strategy comprises a 100,000-iteration autoencoder pre-training phase followed by 1,000,000 iterations for the diffusion bridge. In practice, the Iterative strategy is *not* slower; it is slightly *faster* in wall-clock time per epoch than the End-to-End because each alternating phase updates fewer parameters and therefore requires fewer gradient computations per iteration. Overall runtime for Two-Step is comparable to Iterative.

### A.3.5 Degregraion of sampling steps

we conducted experiments on the ShapeNet dataset using different sampling step counts, starting from 40 and reducing down to 10. We include the results in the table below. We also experimented with more than 40 steps, but this didn't lead to noticeable improvements. Interestingly, reducing the number of steps to 30 appears to be beneficial; however, below 30 steps, the sample quality begins to degrade significantly, as expected.

Model / steps 40 30 20 10 Our 1-NNA ↓ IoU ↑ Table: Results across different step counts. Values are mean std.

### A.3.6 Encoder Quality Ablation: Impact on Latent Bridging

The bridge model assumes that source and target modalities share a sufficiently expressive and consistent latent space. To probe this assumption, we ablate the *encoder/decoder* quality within the Super-Resolution benchmark (FFHQ→CelebA-HQ, $16\times16 \rightarrow 128\times128$). We compare three variants: **(1)** a pre-trained autoencoder (AE) from Latent Diffusion Models [51], **(2)** the same AE architecture trained from scratch (no pre-training), and **(3)** a lightweight 5-layer convolutional encoder/decoder ("Simple CNN"). All other components and training budgets are held fixed.

**Findings.** Table 11 shows that (1) and (2) yield comparable PSNR, indicating limited benefit from pre-training in this setup. In contrast, (3) substantially underperforms, suggesting that architectural capacity and representation quality are critical for robust latent bridging under our losses (bridge, predictive, contrastive).

| Method / Metric | PSNR ↑ |
|---|---|
| (1) – Pre-trained AE | **25.6 ± 0.4** |
| (2) – Vanilla AE | 25.3 ± 0.5 |
| (3) – Simple CNN | 22.9 ± 0.9 |

Table 11: Encoder quality ablation on Super-Resolution (FFHQ→CelebA-HQ). Values are mean ± std over three runs.

### A.3.7 Standard Deviation

For all quantitative experiments, we run the evaluation protocol three times and report the mean and standard deviation of the results. The full results are provided below.

### A.3.8 Qualitative Comparison on SR Task

We present a qualitative comparison of various methods for the super-resolution (SR) task in Fig. 11. The figure shows a set of low-resolution input images and their corresponding high-resolution outputs

Table 12: Complete Results for Tab.4

| Component | ShapeNet | | nuScene | | CelebA-HQ | | |
| | IoU ↑ | 1-NNA ↓ | IoU ↑ | 1-NNA ↓ | PSNR ↑ | SSIM ↑ | LPIPS ↓ |
|---|---|---|---|---|---|---|---|
| (1) U-Net | .635 ± .002 | .518 ± .008 | .216 ± .002 | .818 ± .007 | 23.2 ± 0.3 | 0.57 ± .02 | 0.42 ± .04 |
| (2) DiT | .613 ± .006 | .548 ± .012 | .208 ± .004 | .825 ± .005 | 22.2 ± 0.2 | 0.52 ± .01 | 0.49 ± .03 |
| (3) + Encoder-Decoder | .651 ± .002 | .518 ± .004 | .217 ± .002 | .821 ± .006 | 23.4 ± 0.4 | 0.53 ± .03 | 0.38 ± .01 |
| (4) + Spatial Embedding | .658 ± .003 | .522 ± .007 | .224 ± .004 | .812 ± .007 | 22.9 ± 0.2 | 0.56 ± .05 | 0.41 ± .03 |
| (5) + Learnable Token (Ours) | **.664 ± .002** | **.508 ± .005** | **.233 ± .005** | **.807 ± .008** | **25.6 ± 0.4** | **0.68 ± .03** | **0.32 ± .01** |

generated by four different models: SiT, DiT, EDM, and our proposed method. Each row corresponds to a different model, with the first row displaying the low-resolution inputs and the subsequent rows showing the results generated by the different methods. We can see that the images generated by our method appear more realistic, with fewer artifacts and distortions.

### A.3.9    Qualitative Comparison on multi-view to 3D shape task

We present a qualitative comparison of various methods for the 3D shape generation task from multi-view images Fig. 12. The figure shows a set of two multi-view input images and their corresponding 3D shape outputs generated by four different models: SiT, DiT, EDM, and our proposed method. Each row corresponds to a different model, with the first row displaying the multi-view inputs and the subsequent rows showing the results generated by the different methods. We can see that the images generated by our method appear more realistic, with fewer artifacts and distortions.

### A.3.10    Additional Qualitative Results

We provide additional qualitative results for the super-resolution task in Fig. 13, and for the multi-view-to-3D shape generation, including the ground truth occupancy 3D shape, in Fig. 14. It is important to note that while the generated shapes may not retain the exact same structure, they should exhibit similar features.

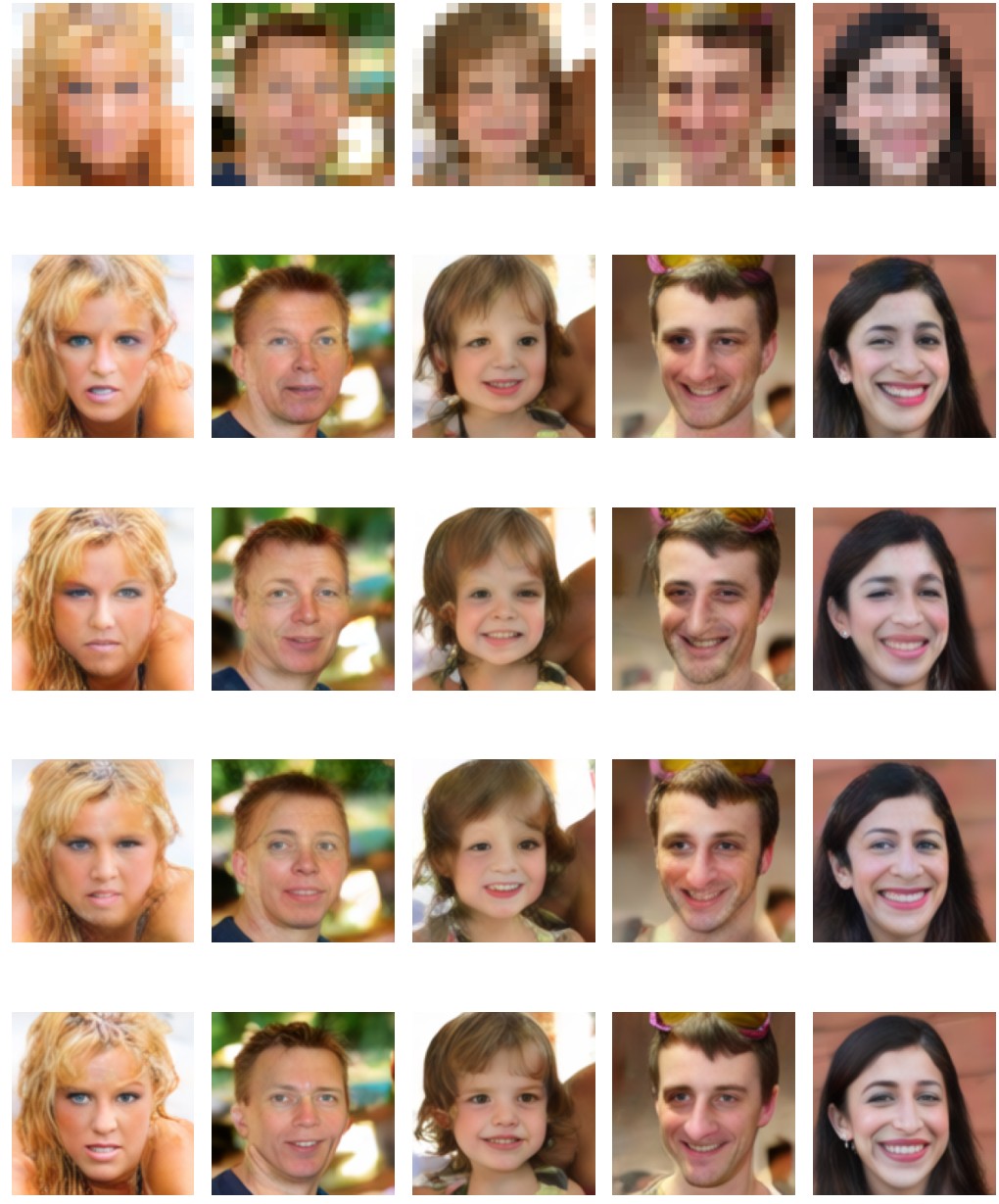

Figure 11: A comparison of all methods on the super-resolution task. The first row displays the low-resolution input images, and the following rows show the high-resolution outputs generated by SiT (second row), DiT (third row), EDM (fourth row), and *LDDBM* (fifth row).

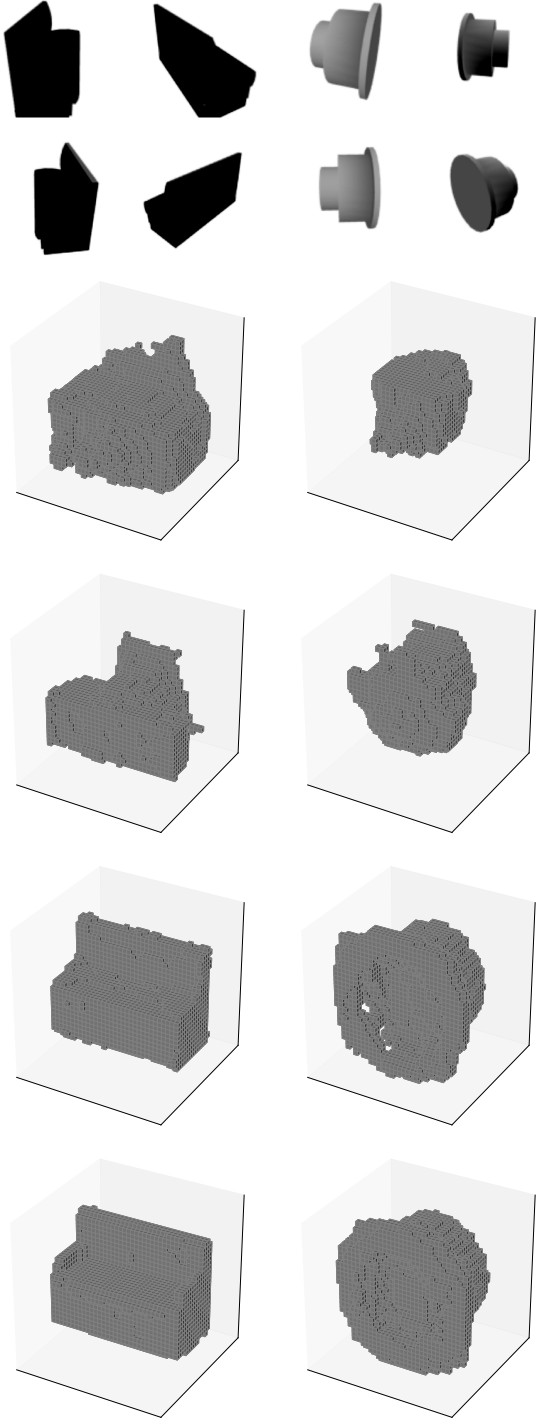

Figure 12: A comparison of all methods on the multi-view to 3D shape task. The first row displays the multi-view input images, and the following rows show the 3D shape outputs generated by SiT (second row), DiT (third row), EDM (fourth row), and *LDDBM* (fifth row).

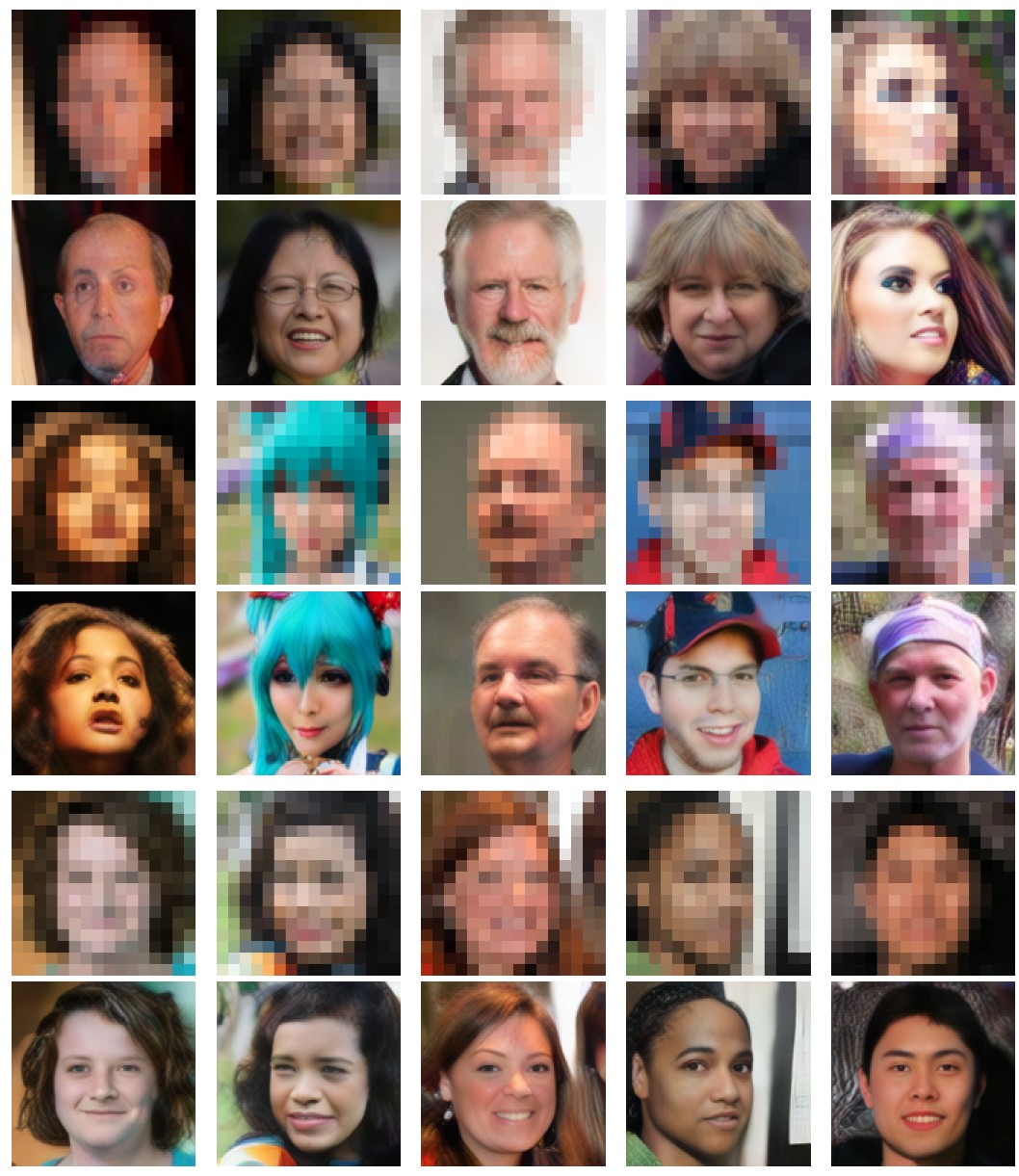

Figure 13: Additional Examples for super resolution task of our model.

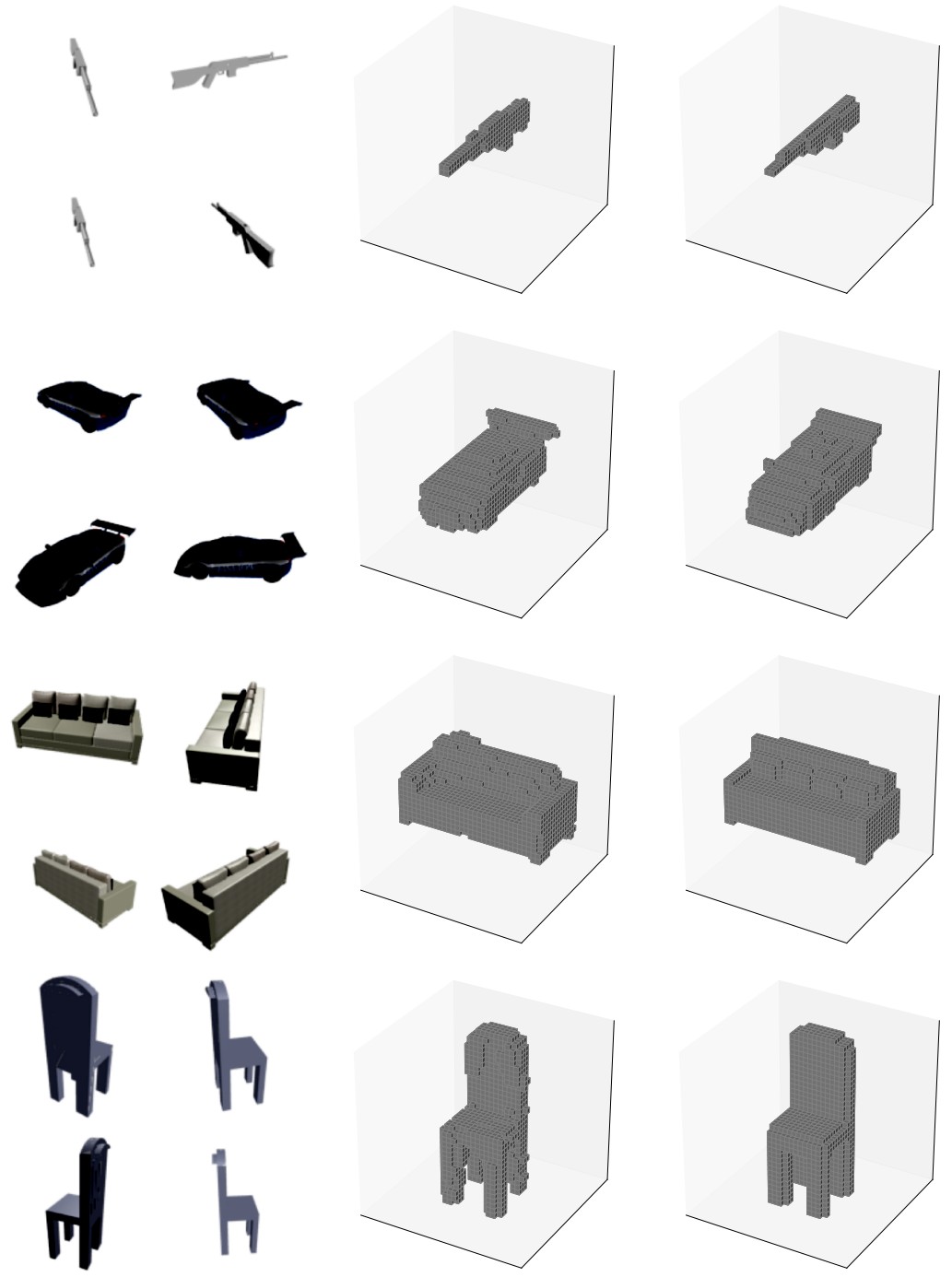

Figure 14: Additional Examples for multi-view to 3D task of our model. Left: the multi-views. Middle: the predicted 3D shape. Right: The ground truth 3D shape.

