# OpenReview forum: "Towards General Modality Translation with Contrastive and Predictive Latent Diffusion Bridge"
_NeurIPS.cc/2025/Conference — NeurIPS 2025 poster_

### Official Review · Reviewer_BFU8 · 2025-06-28

**Clarity:** 3
**Significance:** 3
**Originality:** 3
**Rating:** 4
**Confidence:** 4

**Summary:**

This paper proposes a novel architecture for modality translation, enabling information transfer across heterogeneous modalities.
The key idea is to extend Denoising Diffusion Bridge Models (DDBM) to handle general modality pairs with differing dimensions.
To achieve this, the authors introduce an encoder–decoder framework that projects diverse modalities into a shared latent space. Additionally, a contrastive loss is employed to enforce semantic consistency across domains.
Experiments are conducted on image-to-3D generation and image super-resolution using relatively simple datasets, demonstrating reasonable performance in both tasks.

**Questions:**

1. ***Originality***: This work extends DDBM into latent space using an encoder-decoder, which has been implemented in DPBridge. What differentiates the proposed method from this highly relevant prior work, DPBridge [21]?
2. ***Significance***: The second contribution is the contrastive loss that enhances semantic coherence across domains. The paper only shows visual results in Figure 1 using t-SNE. However, the t-SNE is highly sensitive to the hyperparameters. Do they use the same parameter for both basic and ours results? The paper would be stronger to provide ablations on the contrastive loss in Table 2.
3. ***Clarity***: In figure 1, what's the meaning of the different colors? The denosied output $\hat{z}_t$ still has noise. How to train the encoder-decoder and diffusion denoiser together in eq.8?
4. ***Results***: Similar to the DPBridge, the paper would be stronger to compare with the state-of-the-art methods on each task. For example, for image-to-3D, 3DiM [r1] is also a based model trained from scratch.

[r1] Watson, D., Chan, W., Martin-Brualla, R., Ho, J., Tagliasacchi, A., & Norouzi, M. (2022). Novel view synthesis with diffusion models. arXiv preprint arXiv:2210.04628.

**Ethical Concerns:**

["NO or VERY MINOR ethics concerns only"]

**Final Justification:**

Thanks for the detailed rebuttals. All my concerns have been addressed,  especially the motivation of comparing with DPbridge and the results in other additional tables. Hence, I would like to increase my score and vote to accept this paper.

**Limitations:**

Limitations and potential negative societal impact are not discussed.

**Quality:**

3

**Strengths And Weaknesses:**

### ***Strengths***

* A good attempt at an interesting problem
   * The task of learning a shared latent space for modality translation tasks is very valuable given the heavily different dimensions and distributions in diverse modalities.
   * This paper makes a good initial attempt to tackle this problem in simple cases (simple synthetic ShapeNet dataset)

* Good writing
   * The paper is well written, with clear motivations and illustrative visualizations.

### ***Weaknesses***

* (Major) Significance is not well demonstrated
   * The key motivation of this paper is to extend DDBM to handle general modality pairs with differing dimensions. To achieve this, it is natural to extend it into latent space using an encoder-decoder architecture. However, a prior work DPBridge [21] has implemented a similar idea and shows the best performance on multiple tasks, even compared to these state-of-the-art methods in each domain. Hence, what differentiates the proposed method from this DPBridge [21]?
   * The second contribution is to implement contrastive loss to enhance semantic coherence across domains. However, such a key contribution is only validated in Figure 1. Why is such coherence distribution important? The significance is not well demonstrated in the paper. It would be stronger to ablate it in Table 2.

* (Major) Oversold technical novelty
   * In Line 236, the authors claimed "Inspired by ... we design a denoiser that is better ...". However, the results of DiT on all tasks are worse than UNet in Table 2. What's the benefit of the new architecture?
  * In Line 244, "unlike the original UNet..., our model consumes this memory using the cross-attention...". Such an operation has been employed in DPBridge [21].
  * In Line 210, "Unlike cycle-consistency losses..., our formulation does not require convertibility or backward mapping...". The cycle-consistency losses are designed for unpaired data. However, the proposed architecture is trained with paired data.

* (Minor) Experiments are too simple
    * To explore the new architecture, it is reasonable to deal experiments with Toy datasets. However, it is still important to report the state-of-the-art performance in each domain. For example, for multiview-to-3D shape generation, 3DiM[r1] and Shap-e[r2] are also trained from scratch. The paper would be stronger to compare one of these state-of-the-art approaches for each task.
    * For visual results in the main paper and supp., the fair comparison to prior works is missing.

[r1] Watson, D., Chan, W., Martin-Brualla, R., Ho, J., Tagliasacchi, A., & Norouzi, M. (2022). Novel view synthesis with diffusion models. arXiv preprint arXiv:2210.04628.

[r2] Jun, H., & Nichol, A. (2023). Shap-e: Generating conditional 3d implicit functions. arXiv preprint arXiv:2305.02463.

---

> ### Author Rebuttal · Authors · 2025-07-30
>
> We sincerely thank the reviewer for their thoughtful feedback and for acknowledging the significance of our problem, our initial efforts, and the clear writing of the paper. We have addressed each question and concern raised below and are happy to provide any additional clarification if needed. We apologize for abbreviating some of the reviewer comment references due to space constraints in the rebuttal.
>
> > The key motivation of this paper is to extend DDBM to handle general modality pairs with differing dimensions. To achieve this, it is natural to extend it into latent space using an encoder-decoder architecture. However, a prior work DPBridge \[21] has implemented a similar idea and shows the best performance on multiple tasks, even compared to these state-of-the-art methods in each domain. Hence, what differentiates the proposed method from this DPBridge \[21]?
>
> Thank you for the opportunity to clarify the distinctions between our framework and DPBridge and improve our work's clarity. We will incorporate the below discussion in the new revision's related work.
>
> It is important to note that DPBridge is not a *general* multimodal translation framework under our problem definition in Section 4.1. It was specifically designed for image-to-image translation and is based on the Stable Diffusion (SD) model, which is restricted to the image domain. As such, it cannot be applied to the broader range of tasks we address, including 3D voxel prediction and our newly introduced audio-to-image and image-to-audio translation tasks (see our first response to Reviewer 8Cc5 for more details). Consequently, a direct comparison is not feasible. Beyond this, our method introduces several fundamental differences. First, we propose a composition of loss functions, combining InfoNCE (contrastive), predictive, and standard diffusion bridge losses, and we conduct a thorough ablation study to evaluate their individual and joint effects (see Table 6 in the appendix). DPBridge, in contrast, does not explore alignment-based losses. Second, DPBridge employs a pre-trained Stable Diffusion architecture as its backbone, which limits its generality. In contrast, we design and validate a modality-agnostic architecture specifically tailored to support diverse modality pairs without sacrificing performance. Additionally, while DPBridge relies solely on fine-tuning SD (with all other components frozen), our framework trains all components from scratch, and we are the first to explore various training strategies for building a general-purpose multimodal bridge model, extending the scope of our work beyond architectural and loss design. Finally, it is worth noting that SD v2.1, which DPBridge relies on, was pre-trained on hundreds of millions of images. As shown in their ablation study (Table 3 in their paper), their method performance drops significantly without the use of this foundational model, falling well below state-of-the-art levels.
>
> >  The second contribution is to implement contrastive loss to enhance semantic coherence across domains...
>
> Thank you for raising this important point and helping us improve the paper quality. We do include such an extensive evaluation in our work, as shown in Table 6. However, we appreciate you bringing to our attention that it was not clearly referenced from the main paper. To address this, we have revised the final submission to move this table into the main paper, taking advantage of the additional space now permitted and in accordance with your suggestion. In this evaluation, we present a quantitative comparison of the four loss combinations discussed in the paper. The results show that incorporating the InfoNCE contrastive loss into the training protocol significantly enhances model performance.
>
> > In Line 236, the authors claimed "Inspired by ... we design a denoiser that is better ...". However, the results of DiT on all tasks are worse than UNet in Table 2. What's the benefit of the new architecture?
>
> We apologize for the lack of clarity in our original explanation and thank you for helping us improve the presentation of our contribution. We would like to emphasize that the denoiser architecture used in our framework is **not** a DiT. Instead, we propose an improved architecture tailored for general modality translation tasks, drawing inspiration from natural language translation. Specifically, we employ an encoder-decoder Transformer architecture as our denoising network—unlike DiT, which is based on a decoder-only Transformer design. In addition to this architectural shift, we integrate several components specifically designed to enhance performance in the multimodal setting such as spatial embedding and mask token. As demonstrated in our ablation study (Table 2), this architecture significantly outperforms the commonly used U-Net backbone. Moreover, Table 2 also shows that our denoiser surpasses a standard DiT in modality translation tasks. We have revised Section 4.5 of the paper to include this clarification, in light of this valuable feedback.
>
> > In Line 244, "unlike the original UNet..., our model consumes this memory using the cross-attention...". Such an operation has been employed in DPBridge \[21].
>
> In the DPBridge framework, the authors do not use the cross-attention feature of the UNet, which is reserved for text-prompt conditioning in stable diffusion. As the authors of DPBridge discuss in their paper, "As DPBridge is an image-conditioned generative framework, we employ null-text embedding for all cross-attention layers to eliminate the impact of text prompts," suggesting that the application of any cross-attention layer in their framework has been nullified. The target modality information is concatenated with the source modality at the input of the UNet, along the channels dimension. Contrastingly, since our model uses a transformer encoder and decoder, we use cross-attention to condition our modality translation.
>
> > In Line 210, "Unlike cycle-consistency losses..., our formulation does not require convertibility or backward mapping...". The cycle-consistency losses are designed for unpaired data. However, the proposed architecture is trained with paired data.
>
> We would like to clarify that our intention was simply to convey that our method does not require a full cycle starting from the diffusion bridge and returning back. This was mentioned as a minor comment to help provide additional intuition for the reader, and was not intended as a comparison to unpaired data methods. We understand how this phrasing may have caused confusion. To address the reviewer’s concern and avoid any further misunderstanding, we will remove this line in the revised version of the paper.
>
> >  To explore the new architecture, it is reasonable to ...
>
> We appreciate the opportunity to clarify our position on this point. While we agree that comparisons to state-of-the-art methods can be informative, we would like to emphasize that such comparisons are not entirely fair to our framework. Our model is designed to be general and capable of handling a wide range of multimodal translation tasks. In contrast, state-of-the-art methods in the benchmarks we evaluate on are highly specialized, incorporating strong inductive biases tailored to the specific data domain through architectural choices, loss functions, and training procedures.
>
> Nonetheless, we agree that including such comparisons can help convey the broader context and performance trade-offs. Therefore, in response to your suggestion, we have added the results of a task-specific state-of-the-art method \[1] to the quantitative evaluation table for the Multiview-to-3D Shape Generation task (highlighted in gray). Additionally, for the Zero-Shot Low-to-High Resolution Generation task, we already include a comparison with DiWa. Finally, we have also added comparisons to relevant baselines for the new image-to-audio and audio-to-image tasks introduced in this work. Thank you again for raising this point and helping us improve the completeness of our benchmarking.
>
> \[1] Dan Wang, et al - Multi-view 3D Reconstruction with Transformer
>
> > For visual results in the main paper and supp., the fair comparison to prior works is missing.
>
> We include a visual comparison in Figures 7 and 8 of the appendix. However, we acknowledge that there was no clear reference to this experiment in the main paper. To improve the clarity and reading flow, we have added a reference to this experiment in the revised version of the main text. Thank you for pointing this out.
>
> > This work extends DDBM ...
>
> Please see our first comment.
>
> > The second contribution is the contrastive loss ...
>
> Thank you for this question. Yes, we use the exactly same hyperparameters for both the basic and ours. Regarding the second point, we include ablations in the appendix; please see our second comment for full explanation.
>
> > In figure 1, what's the meaning of the different colors? The denoised output still has noise. How to train the encoder-decoder and diffusion denoiser together in eq.8?
>
> Each color represents a category in the dataset—for example, a car or an airplane. We compute both the reconstruction loss and the predictive loss using the reconstructions of the original latent representation and the denoised latent representation. This setup enables standard backpropagation-based training using a single optimizer, as all components are implemented within a unified PyTorch module.
>
> > Results: Similar to the DPBridge, the paper would be ...
>
> As previously addressed, we welcomed the suggestion and have added comparisons with state-of-the-art methods. Unlike DPBridge, which is specifically designed for the tasks it addresses and leverages a powerful pre-trained foundational model for fine-tuning, our framework is intended to be much more general, and thus a direct comparison may be unfair. As discussed earlier, this introduces a natural trade-off between performance and generality.

---

> > ### Comment · Reviewer_BFU8 · 2025-08-06
> > **reply to the authors**
> >
> > Thank the author providing these detailed rebuttals. All my concerns have been addressed. I have not noticed the training difference between the DPbridge and the proposed method, the authors' reply is reasonable and this should be useful for the community. Hence, I would llike to increase my score and vote to accept for this work.

---

> > > ### Author Response · Authors · 2025-08-06
> > > **Thank You**
> > >
> > > Thank you for helping to improve the work through your thoughtful and constructive feedback.

---

### Official Review · Reviewer_8Cc5 · 2025-07-02

**Clarity:** 3
**Significance:** 3
**Originality:** 2
**Rating:** 5
**Confidence:** 3

**Summary:**

This work extends Denoising Diffusion Bridge Models (DDBMs) for general modality translation (MT) via a shared latent space leveraging modality-specific encoder/decoders. It utilizes a combination of bridge loss, predictive loss, and contrastive loss for accurate and high-fidelity domain translation. The proposed approach is evaluated on multi-view to 3D shape generation, image super-resolution, and multi-view scene synthesis tasks, showcasing improved performances.

**Questions:**

- The iterative training protocol significantly improves performance, however, it would be useful to provide more specific details. Is the training time for the iterative procedure significantly longer compared to two-step or end-to-end?
- It would be great if the authors could provide more intuition for the MASK token and some analysis for why its inclusion drastically improves performance on some tasks (+2.7 points on CelebA-HQ PSNR).
- In the iterative process, are all three losses applied for both reconstruction and bridge alignment steps?
- It would be great if the specific DDBM noising method is provided in the background section or appendix for readers who are unfamiliar.

Post rebuttal note:
The rebuttal has clarified most of my concerns and I will increase my rating to reflect this.

**Ethical Concerns:**

["NO or VERY MINOR ethics concerns only"]

**Final Justification:**

The rebuttal has clarified most of my concerns and I will increase my rating to reflect this.

**Limitations:**

Yes - computational tradeoffs as well as the limitation of training on paired data are mentioned.

**Paper Formatting Concerns:**

None.

**Quality:**

3

**Strengths And Weaknesses:**

Strengths:
- The key contribution of a modality-agnostic translation method is novel, design choices such as the encoder-decoder architecture for the bridge model are well-motivated.
- Comprehensive ablations are provided for architectural and loss components as well as training procedures for all benchmarks.
- The proposed approach is effective across all benchmarks.

Weaknesses:
- The proposed approach is evaluated primarily on vision-based modality translation tasks, it would be interesting to see how well this approach works outside of vision (e.g., text to image).
- The bridge model requires that the source and domain share a robust latent space. Ablations with lower quality encoder/decoders for each modality would elucidate the effectiveness of the proposed losses.
- The image super-resolution experiments focus exclusively on faces. It would be interesting to evaluate how well the proposed pipeline generalizes to other domains such as natural images.

---

> ### Author Rebuttal · Authors · 2025-07-30
>
> We thank the reviewer for the thoughtful feedback and for recognizing the novelty of our framework design and the extensiveness of our experiments. Below, we address the questions and concerns raised. We are happy to provide any additional clarification if needed.
>
> > The proposed approach is evaluated primarily on vision-based modality translation tasks, it would be interesting to see how well this approach works outside of vision (e.g., text to image).
>
> To directly address this concern, which was also raised by another reviewer, diversify our benchmarking, and strengthen the robustness of our method results, we introduce two new tasks to our model: *Image-to-Audio* and *Audio-to-Image*, following the benchmark proposed in \[1]. In these tasks, a multimodal translation model receives one modality as input and generates a meaningful representation, which is then used for downstream classification.
>
> We evaluate our method against the SiT variant, which showed the overall second-best performance on other benchmarks, as well as against the current state-of-the-art, which is a task-specific method tailored for audio and image representations. The results are shown in the table below. For implementation and evaluation details, we refer the reader to \[1], and we have also added full experimental details, including encoder and decoder architectures, in the appendix of the revised version.
>
> As shown in the table—where higher values indicate better classification accuracy—our method outperforms SiT. However, a performance gap remains when compared to the specialized method in \[1], which is expected given its very task-specific nature. Overall, the results below demonstrate the generality of our framework, its superiority among general multimodal translation models and its possible adaptation to more diversified domains.
>
> | **Method** | **Face to Voice** &uarr; | **Voice to Face**  &uarr;|
> | :--------- | :---------------- | :---------------- |
> | Our method | 71.2              | 75.1              |
> | SiT        | 65.7              | 68.3              |
> | \[1]       | 79.5              | 81.0              |
>
> *Table: Performance comparison on cross-modal face-voice tasks.*
>
> \[1] Seeing Voices and Hearing Faces: Cross-modal biometric matching - A. Nagrani, S. Albanie, A. Zisserman
>
> > The bridge model requires that the source and domain share a robust latent space. Ablations with lower quality encoder/decoders for each modality would elucidate the effectiveness of the proposed losses.
>
> Thank you for raising this ablation perspective, which helps improve the understanding of our method. To address this point, we conducted an experiment within the Super-Resolution benchmark evaluating the impact of encoder architecture. Specifically, we examined three variants:
>     (1) a pre-trained autoencoder \[1],
>     (2) the same encoder architecture without pre-training, and
>     (3) a simpler encoder consisting of a 5-layer convolutional neural network.
>     We report the results in the table below and will include this experiment in the final revision. Briefly, we observe that both (1) and (2) achieve comparable PSNR scores, suggesting that pre-training provides only marginal benefit in this case. However, variant (3) shows a noticeable drop in performance, indicating that architectural capacity and representation quality play a crucial role in the final outcome. This experiment underscores the importance of encoder quality in our framework.
>
> | **Method / Metric** | **PSNR** &uarr; |
> | :------------------ | :------------- |
> | (1) – Pre-trained AE | $25.6 \pm 0.4$ |
> | (2) – Vanilla AE    | $25.3 \pm 0.5$ |
> | (3) – Simple CNN    | $22.9 \pm 0.9$ |
>
> *Table: Comparison of PSNR across different encoder architectures. Values are mean $\pm$ std.*
>
> \[1] R. Rombach, A. Blattmann, D. Lorenz, P. Esser, and B. Ommer. High-resolution image synthesis with latent diffusion models
>
> > The image super-resolution experiments focus exclusively on faces. It would be interesting to evaluate how well the proposed pipeline generalizes to other domains such as natural images.
>
> We incorporated the **DIV2K dataset**—a large, diverse collection of high-quality RGB images, into our framework and extracted visual results. It is encouraging to observe that our model successfully performs the super-resolution task, generating high-quality outputs on this dataset. For quantitative evaluation, our method outperforms DiWa, achieving a PSNR score of 28.9 compared to DiWa’s 28.2. These results will be included in the final revision of the paper.
>
> > The iterative training protocol significantly improves performance, however, it would be useful to provide more specific details. Is the training time for the iterative procedure significantly longer compared to two-step or end-to-end?
>
> Thank you for raising this important point. For both the End-to-End and Iterative strategies, we used the same total number of training iterations (1,000,000). In practice, we observed that the Iterative strategy is faster in terms of wall-clock time, as it involves fewer gradient computations per iteration. Regarding the Two-Step strategy, we first trained the autoencoders for 100,000 iterations and then trained the diffusion bridge for an additional 1,000,000 iterations. The overall runtime of this variant is comparable to that of the Iterative approach. We will include this discussion and the corresponding runtime comparisons in the revised version of the paper. Thank you again for bringing this to our attention and helping us improve the clarity of our work.
>
> > It would be great if the authors could provide more intuition for the MASK token and some analysis for why its inclusion drastically improves performance on some tasks (+2.7 points on CelebA-HQ PSNR).
>
> The MASK token, also known as the context summarization token, is a token added to a lot of common transformer-based architectures that participates in all the attention mechanisms, even through the causal ones. Intuitively, as the name suggests, this token learns to effectively combine all the information from the tokens in the context window and generates a *summary* embedding token. This token is commonly used when your application needs to use a summary of all the tokens in the context window, such as when using a ViT for classification applications, they pad with a \[CLS] token that generates a summary embedding that can be fed into any downstream classifier model. In our method, we use this token to summarize the semantics stored in one of the modalities to effectively transfer them into the other modality using our technique. We have incorporated the above discussion in the final revision to help clarify the motivation.
>
> > In the iterative process, are all three losses applied for both reconstruction and bridge alignment steps?
>
> That's a great question. During training, we compute all three losses, but only propagate gradients through the architectural components that are currently being trained. Specifically, when training the encoders and decoders, the bridge parameters remain frozen; conversely, when training the bridge, the encoder and decoder parameters are not updated.
>
> > It would be great if the specific DDBM noising method is provided in the background section or appendix for readers who are unfamiliar.*
>
> We will be happy to make this information easier to find in the final revision and elaborate more on the topic in the appendix. The specific DDBM noising and denoising processes are described in Section 3, including the forward SDE in Eq. (1), the reverse-time formulation in Eq. (2), and the corresponding training loss in Eq. (3). We also clarify that we adopt the variance exploding (VE) noise schedule from \[68] (lines 118–120). However, we will try to include more information that will help the reader to gain a better understanding of the DDBM framework.

---

> > ### Comment · Reviewer_8Cc5 · 2025-08-05
> >
> > The rebuttal has clarified most of my concerns and I will increase my rating to reflect this.

---

> > > ### Author Response · Authors · 2025-08-06
> > > **Thank You**
> > >
> > > Thank you for your constructive and valuable feedback, which helped move the work forward.

---

### Official Review · Reviewer_TLPh · 2025-07-03

**Clarity:** 3
**Significance:** 2
**Originality:** 3
**Rating:** 3
**Confidence:** 3

**Summary:**

This paper introduces a general-purpose framework for Modality Translation (MT) using a novel extension of Denoising Diffusion Bridge Models (DDBMs) into a shared latent space. It proposes a domain-agnostic encoder-decoder architecture and augment it with predictive and contrastive losses to improve semantic consistency and translation accuracy across modalities of differing dimensionality. The proposed method is evaluated on multiple tasks including multi-view to 3D shape generation, image super-resolution, and multi-view scene synthesis, and reaches leading performance over baselines.

**Questions:**

- Is the bridge model $B$ in the predictive loss $L_{pred}$ also being trained under the bridge loss $L_{bridge}$, or is it frozen when $L_{pred}$ is backpropogating?

- In terms of transitions between different modalities using diffusion models, how does the proposed method relate to and differ from CrossFlow[1] or FlowTok[2], which both directly map between images and texts?

[1] Flowing from Words to Pixels: A Noise-Free Framework for Cross-Modality Evolution
[2] FlowTok: Flowing Seamlessly Across Text and Image Tokens

**Ethical Concerns:**

["NO or VERY MINOR ethics concerns only"]

**Final Justification:**

Thanks the authors for the informative feedback. I think for the additional experiments on non-visual modalities, I'd further suggest that 1) text would be a more popular and convincing modality to evaluate, and 2) other baseline bridge models need to be compared to demonstrate the effect of the proposed modules, besides the provided ordinary models -- especially [1] is dated back to 2018.

Overall I think there are still potential improvements for this paper. But I wouldn't object to the acceptance to it, given the feedback from other reviewers.

**Limitations:**

The paper only very briefly mentions the computational limitations of processing higher-resolution images.

**Paper Formatting Concerns:**

No formatting concerns found.

**Quality:**

2

**Strengths And Weaknesses:**

Strengths:

- The proposed method novelly designs an end-to-end predictive loss to train the autoencoder, and proposes to iteratively train the reconstruction and bridging models, which are different from current mainstream two-stage visual generative models.

Weaknesses:

- [Major] The paper targets to address generic transition tasks between different modalities, and the proposed method, including the training scheme and model architecture, is claimed to be designed for this purpose. However, the benchmark tasks are relatively simple, and especially, don't involve much different modalities but are mostly limited to pixel data like images/masks and voxels, which are still very similar. There is no clue that distinct modalities like language, audio, action, or other unusual sensor data formats are involved. Therefore the capability of the proposed framework is doubted.

- The paper writing could be improved. Specifically, there are too many generally descriptive words (e.g. what a component is designed for) of the proposed method, but without actually illustrating what it really is (e..g how a component is designed), since the abstract, introduction and until Ln 206. Concrete definitions of the proposed method should be summarized and emphasized even before the method section, with various granularity.

- The detailed model architecture related to Sec. 4.5 needs to be presented in the main paper. Many descriptive words could be more concise and concentrated, and the analyses (Ln191-203) and even the background (Sec. 3) could be moved to the Appendix if necessary.

---

> ### Author Rebuttal · Authors · 2025-07-30
>
> We thank the reviewer for the constructive feedback and for acknowledging the novelty of our framework. Below, we respond to the questions and concerns raised. We are happy to provide any additional clarification if needed.
>
> > [Major] The paper targets to address generic transition tasks between different modalities, and the proposed method, including the training scheme and model architecture, is claimed to be designed for this purpose. However, the benchmark tasks are relatively simple, and especially, don't involve much different modalities but are mostly limited to pixel data like images/masks and voxels, which are still very similar. There is no clue that distinct modalities like language, audio, action, or other unusual sensor data formats are involved. Therefore the capability of the proposed framework is doubted.
>
> To directly address this concern, which was also raised by another reviewer, diversify our benchmarking, and strengthen the robustness of our method results, we introduce two new tasks to our model: *Image-to-Audio* and *Audio-to-Image*, following the benchmark proposed in \[1]. In these tasks, a multimodal translation model receives one modality as input and generates a meaningful representation, which is then used for downstream classification.
>
> We evaluate our method against the SiT variant, which showed the overall second-best performance on other benchmarks, as well as against the current state-of-the-art, which is a task-specific method tailored for audio and image representations. The results are shown in the table below. For implementation and evaluation details, we refer the reader to \[1], and we have also added full experimental details, including encoder and decoder architectures, in the appendix of the revised version.
>
> As shown in the table, where higher values indicate better classification accuracy, our method outperforms SiT. However, a performance gap remains when compared to the specialized method in \[1], which is expected given its very task-specific nature. Overall, the results below demonstrate the generality of our framework, its superiority among general multimodal translation models and its possible adaptation to more diversified domains.
>
> | **Method** | **Face to Voice** &uarr; | **Voice to Face**, &uarr; |
> | :--------- | :---------------- | :---------------- |
> | Our method | 71.2              | 75.1              |
> | SiT        | 65.7              | 68.3              |
> | \[1]       | 79.5              | 81.0              |
>
> *Table: Performance comparison on cross-modal face-voice tasks.*
>
> \[1] Seeing Voices and Hearing Faces: Cross-modal biometric matching - A. Nagrani, S. Albanie, A. Zisserman
>
>
> > The paper writing could be improved. Specifically, there are too many generally descriptive words (e.g. what a component is designed for) of the proposed method, but without actually illustrating what it really is (e.g. how a component is designed), since the abstract, introduction and until Ln 206. Concrete definitions of the proposed method should be summarized and emphasized even before the method section, with various gra
>
> While we provide formal definitions of our method starting in Section 4 and Appendix A.2 (e.g., latent bridge in Eq. 4, full objective in Eq. 10), we agree that an earlier, more concrete summary would improve clarity. We have changed the paper structure to include a brief structured overview of the method (summarizing key components and pointing to Figure 2) earlier in the paper to better guide the reader through the technical core and directly explain components. We are more than happy to include any further clarification and restructuring if needed.
>
> > The detailed model architecture related to Sec. 4.5 needs to be presented in the main paper. Many descriptive words could be more concise and concentrated, and the analyses (Ln191-203) and even the background (Sec. 3) could be moved to the Appendix if necessary.
>
> Thank you for pointing out this point, which will help improve the clarity of our paper. Initially, due to space constraints, we omitted the architecture figure related to Section 4.5. However, in light of your comment—and given that the revised submission allows for an additional page—we have restructured the paper to include this figure, which we agree significantly enhances clarity.
>
>
> > Is the bridge model in the predictive loss also being trained under the bridge loss, or is it frozen when is backpropagating?*
>
> It depends on the training strategy. In both the Two-Step and End-to-End strategies, the model is always actively trained throughout the training process (With a small exception in the frist step of the Two-Step where only the encoders/decoers are trained). In the Iterative strategy, the model remains frozen if and only if the encoders are being trained, which occurs in an alternating fashion.
>
> > In terms of transitions between different modalities using diffusion models, how does the proposed method relate to and differ from CrossFlow \[1] or FlowTok \[2], which both directly map between images and texts?*
>
> Thank you for raising this point and giving us the opportunity to clarify how the proposed method relates to and differs from our work. We have added the following discussion to the related work in the final revision to better differentiate our model:
>
> The first fundamental difference between our method and the two referenced works lies in their task specificity. Both of the mentioned methods focus exclusively on a particular latent transition: from text to image. They incorporate task-specific components into the architecture and loss design to facilitate this transition. In contrast, our work aims to develop a general-purpose solution for latent bridging. We investigate several general aspects, including training strategies, which we study in depth.
>
> Another major distinction is in the bridge architecture. While both prior works rely on off-the-shelf DiT architectures, our approach introduces a custom-designed architecture specifically tailored to the more general latent bridging setting. We demonstrate both its efficiency and effectiveness in this context.
>
> Finally, from a theoretical standpoint, our method builds on Denoising Diffusion Bridge Models (DDBMs), a fundamentally different framework from the Flow Matching approach used in the referenced works. These frameworks differ in both their loss formulations and sampling procedures. Although Flow Matching is theoretically valid for bridging any two distributions, similar to DDBMs, the specific implementations in the referenced works are designed for Gaussian-to-data transitions. Indeed, both methods incorporate a Gaussian loss term on one side of the bridge, which is not required in our model.
>
> Interestingly, recent theoretical work such as \[3] proposes extensions to Flow Matching that could enable latent bridging without requiring one side to be Gaussian. This presents a compelling direction for future research. However, the current versions of those methods are constrained by this Gaussian constraint, whereas our DDBM-based approach operates without such a limitation.
>
> \[3] Stochastic interpolants with data-dependent couplings - Michael S Albergo\*, Mark Goldstein\*, Nicholas M Boffi, Rajesh Ranganath, Eric Vanden-Eijnden

---

> > ### Comment · Reviewer_TLPh · 2025-08-06
> >
> > Thanks the authors for the informative feedback. I think for the additional experiments on non-visual modalities, I'd further suggest that 1) text would be a more popular and convincing modality to evaluate, and 2) other baseline bridge models need to be compared to demonstrate the effect of the proposed modules, besides the provided ordinary models -- especially [1] is dated back to 2018.
> >
> > Overall I think there are still potential improvements for this paper. But I wouldn't object to the acceptance to it, given the feedback from other reviewers.

---

> ### Author Response · Authors · 2025-08-07
> **Thank You**
>
> Thank you once again for engaging in this discussion and providing valuable feedback. We will take your comments into careful consideration and incorporate them into our future work. We appreciate you pointing these aspects out.

---

### Official Review · Reviewer_Egf9 · 2025-07-07

**Clarity:** 3
**Significance:** 3
**Originality:** 3
**Rating:** 4
**Confidence:** 3

**Summary:**

This paper introduces a novel approach to multi-modal domain translation, specifically addressing a key limitation of existing bridge models. While current methods, such as those relying on optimal transport for tasks like image-to-image or image-to-3D translation, are often restricted by the defined marginal spaces of their start and end distributions, this work offers a new solution to model the distribution translation using latent bridge by estimating a conditional distribution conditioned on both the modalities by factorizing the conditional distribution and introducing a shared latent space.
In addition they define a modality agnostic transformer based encoder -decoder training framework eliminating the need to additional conditioning and any implicit biases added due to the architecture.

**Questions:**

1. The paper demonstrates strong capabilities in multi-modal translation. To further showcase the versatility and robustness of the proposed modality agnostic transformer, suggest exploring its application to outpainting and/or image editing tasks in image-to-image translation? This might provide some insights on spatial relationships and compositional representations?

2. Sampling complexity of DDBMs are generally higher to achieve accurate interpolations can the authors elaborate on how the complexity is addressed ?

3.  While it is impressive that the approach does not require any conditioning, It would be good to see how does the objective affect the training complexity.  Does the process of learning the 'bridge' between two distributions, and the modality-agnostic denoiser in a shared latent space, introduce a higher computational cost (e.g., longer training times, increased GPU memory usage) per iteration or per epoch compared to standard multi-modal EDMs. In additional how does the training convergence compared to standard diffusion pipelines.

**Ethical Concerns:**

["NO or VERY MINOR ethics concerns only"]

**Final Justification:**

I have looked at the authors' response to questions and I retain my score.
1. Training stability:  More empirical ablations are required to understand the training stability over different encoder-decoder architectures as mentioned by the authors
2. The impact of inference vs latency tradeoff is clear.

I will retain the score so far, overall direction is promising and the paper is well motivated.
I have reduced my confidence to 3, I am not an expert/unfamilar in the litterature work the more I read the authors rebuttals.

**Limitations:**

The approach is interesting and very relavent, However I have requested for clarifications on complexity and ability to model spatial relationships.

**Paper Formatting Concerns:**

Minor comment,
The transformer encoder-decoder block and modality specific encoder decoder blocks

**Quality:**

3

**Strengths And Weaknesses:**

Strengths:

1. The paper is well motivated and the structured and directly addresses the limitations of bridge models and modality specific encoder decoder architectures.


2. The authors introduce a sampling approach for training diffusion-based bridge models, specifically by learning a joint latent space shared across multiple modalities. This method offers a latent space that is trained to be more semantic. Current approaches to learn a shared latent space in multi-model models involve learning a bottleneck layer or mixture of expert architectures which require large number of pairs samples to learn from and the validation and practical deployment of these specific diffusion based approaches have been challenging.

3. This work proposes a general purpose transformer based encoder-decoder architecture that is trained for 3 tasks super-resolution, 3D shape generation and occupancy tasks. The idea of universal auto-encoder decoder architecture is novel and to the best of my knowledge not been explored.

4.  This work qualitatively illustrates the limitations of baseline objective demonstrating good intuitions on semantic alignment . In addition to comparing the proposed architecture Distribution Translation DiT against UNET and naive DiT architecture

Weakness:

1.  Although the results look very promising, the paper does not sufficiently address or analyze the inference time complexity of their bridge model approach compared to other multi-modal shared latent space methodologies that define a separate encoder decoder models.   Secondly Sampling complexity of DDBMs are generally higher to achieve accurate interpolations, would be curious to see how is the sample quality affected

2.  DDPMs inherently support compositional representations especially would be beneficial when performing multi-modal generations. The paper did not address ability to preserve and transfer semantic relationships, compositional representations between modalities and retain usefull properties for image tasks.

3. In general the training framework involves iteratively learning the auto-encoder/latent spaces jointly with the bridge essentially learning the marginal distributions making it a harder optimization problem.

---

> ### Author Rebuttal · Authors · 2025-07-30
>
> We thank the reviewer for the constructive feedback and for recognizing the importance of our motivation, the novelty of our approach, and the extensiveness of our evaluation. Below, we address the concerns and questions raised. We'd be happy to provide further clarification on any additional points, if needed.
>
> > Although the results look very promising, the paper does not sufficiently address or analyze the inference time complexity of their bridge model approach compared to other multi-modal shared latent space methodologies that define a separate encoder decoder models. Secondly Sampling complexity of DDBMs are generally higher to achieve accurate interpolations, would be curious to see how is the sample quality affected
>
> Thank you for raising these two points, which help clarify an important aspect of our work.
>
> Regarding the first comment, we've added a **Complexity Analysis** in Section A.3.2 of the appendix, where we compare and contrast the inference-time complexity of our approach with other methods. We also include a memory analysis. The results demonstrate that our model offers a clear advantage in terms of both model size and runtime efficiency. To make this section more accessible to readers, we've added a reference to it in the main text. We appreciate you pointing this out.
>
> To address the second point, we conducted experiments on the ShapeNet dataset using different sampling step counts, starting from 40 and reducing down to 10. We include the results in the table below. We also experimented with more than 40 steps, but this didn't lead to noticeable improvements. Interestingly, reducing the number of steps to 30 appears to be beneficial; however, below 30 steps, the sample quality begins to degrade significantly, as expected.
>
> | **Model / steps** | **40** | **30** | **20** | **10** |
> | :---------------- | :------------ | :------------ | :------------ | :------------ |
> | Our 1-NNA &darr;        | $508 \pm .005$ | $504 \pm .006$ | $522 \pm .008$ | $631 \pm .018$ |
> | IoU     &uarr;          | $664 \pm .002$ | $663 \pm .004$ | $642 \pm .004$ | $533 \pm .014$ |
>
> *Table: Results across different step counts. Values are mean $\pm$ std.*
>
> Thank you again for highlighting this issue. We will include these findings in the final revision.
>
>
> > DDPMs inherently support compositional representations especially would be beneficial when performing multi-modal generations. The paper did not address ability to preserve and transfer semantic relationships, compositional representations between modalities and retain usefull properties for image tasks.
>
> Thank you for raising this important point and helping us improve the quality of our paper. It's worth emphasizing that we intentionally designed our benchmark to focus on modality translation between modalities with different dimensionalities. However, we fully agree that including benchmarking tasks with aligned modality dimensions can further enhance the robustness and generalizability of our method. To address this, we leveraged the **edges-to-bags task from the DDBM benchmark**, which serves as an appropriate evaluation for your suggestion. We tested our framework on this task, analyzing two key aspects: sample quality (FID) and inference performance (runtime in seconds). The results are presented in the table below and will be included in the final revision of the paper.
>
> Briefly, the findings highlight a clear trade-off: while our method remains competitive in terms of quality, it offers more than 2x faster inference speed compared to DDBM.
>
> | **Method / Metric** | **FID** | **Inference Time (sec)** |
> | :------------------ | :------ | :----------------------- |
> | Ours         &darr;       | 4.17    | 7.8                      |
> | DDBM        &darr;        | 2.93    | 16.9                     |
>
> *Table: Comparison of FID and inference time (in seconds).*
>
> Inference time is measured on a batch size of 256. We used [1] autoencoder as the encoder-decoder backbone for our method.
>
> [1] R. Rombach, A. Blattmann, D. Lorenz, P. Esser, and B. Ommer. High-resolution image synthesis with latent diffusion models
>
> > In general the training framework involves iteratively learning the auto-encoder/latent spaces jointly with the bridge essentially learning the marginal distributions making it a harder optimization problem.
>
> We agree that jointly learning both the encoder/decoder (which define the latent space marginals) and the diffusion bridge (which assumes fixed marginals during training) introduces a nontrivial optimization challenge. To address this, we introduced and evaluated the iterative training strategy (see Appendix A.3.4 and Table 7) that alternates between optimizing the autoencoders and the bridge module. This approach reduces interference between components and improves the final performance. For example, our iterative setup achieves a 1-NNA of 0.508 and IoU of 0.664 on ShapeNet, outperforming two-step and fully end-to-end alternatives. Its important to note, that while the iterative setup is highly stable, we do not observe major stability issues in other setups as well.
>
> While our experiments focus on jointly trained encoders, we note that the use of pre-trained modality-specific foundation models—such as CLIP or DINO for vision, and wav2vec or Whisper for audio—could further mitigate this training burden. These models may stabilize the latent marginals early in training and offer performance gains, particularly in low-data regimes. We consider this a promising direction for future work.
>
> > The paper demonstrates strong capabilities in multi-modal translation. To further showcase the versatility and robustness of the proposed modality agnostic transformer, suggest exploring its application to outpainting and/or image editing tasks in image-to-image translation? This might provide some insights on spatial relationships and compositional representations?
>
> Please refer to the discussion in the 2nd response.
>
> > Sampling complexity of DDBMs are generally higher to achieve accurate interpolations can the authors elaborate on how the complexity is addressed ?
>
> The DDBM sampler employs a sampling strategy similar to that used in the Elucidated Diffusion Model (EDM). We observe that even complex bridging tasks, such as the one considered in our work, can be accomplished in no more than **40 sampling steps**—comparable to the number used by EDM for datasets like FFHQ or AFHQv2, which typically require around 40 steps depending on the implementation. Furthermore, the experiment discussed in the first comment provides additional insight, suggesting that further optimization of the number of sampling steps is possible. We'd be happy to provide any further clarification if needed.
>
> > While it is impressive that the approach does not require any conditioning, It would be good to see how does the objective affect the training complexity. Does the process of learning the 'bridge' between two distributions, and the modality-agnostic denoiser in a shared latent space, introduce a higher computational cost (e.g., longer training times, increased GPU memory usage) per iteration or per epoch compared to standard multi-modal EDMs. In additional how does the training convergence compared to standard diffusion pipelines.
>
> Regarding the first point, please refer to our first response; we'd be happy to address any more aspects. As for the second point, we observe that the convergence behavior is very similar. The training time of our model closely matches that of the EDM variants, including both the U-Net and DiT implementations.

---

### Note · Authors · 2025-08-12

Dear AC,

We appreciate the opportunity to provide these final remarks and thank the reviewers for their constructive feedback, which has greatly strengthened our submission. Below, we summarize the main concerns and our responses.

1. Generality beyond vision & task diversity – Reviwers TLPh and 8Cc5 noted our benchmarks were vision-centric. To adress this concern, we added two new cross-modal tasks (Image→Audio, Audio→Image). Our method outperformed the best general-purpose baseline (SiT) and performed competitively with a task-specific SOTA, demonstrating robustness across domains with different dimensionalities. In its final remark TLPh suggested for future work to include text-to-image as additional setup and add more baselines for comparison.

2. Complexity and sampling efficiency – Concerns were raised by Egf9 about inference/training complexity and DDBM sampling cost - we referenced to our
detailed complexity/memory analysis (Appendix A.3.2) showing our model is smaller and faster than comparable baselines. To adress further concern, we conducted sampling studies on ShapeNet, showing quality is maintained down to ~30 steps.

3. Semantic and compositional preservation – Egf9 requested evidence of preserving semantic relationships. We incorporated an edges→bags task, showing competitive quality and superior runtime. We also clarified how our contrastive loss improves alignment, moving ablation results (Table 6) into the main paper.

4.  Writing clarity - following  TLPh comments regarding improved writing, we restructured the paper to introduce a brief method overview earlier, added the detailed architecture diagram to the main text, and clarified distinctions from DiT, U-Net, and other designs (e.g., our encoder-decoder Transformer with mask token).

5. Relation to prior work (DPBridge, CrossFlow, FlowTok) – Following TLPh and BFU8’s concerns and the reviewers’ final feedback, we clarified the key differences between our method and related work, and incorporated this discussion into the final revision.

6. Additional ablations and domain tests – We evaluated encoder quality (pre-trained vs. vanilla vs. simple CNN) and super-resolution on DIV2K (outperforming DiWa), showing generalization beyond faces.

The above summarizes the key discussion points; Other concerns we addressed may be omitted due to space constraints. We once again thank all participants for their valuable feedback, which has significantly improved our work.

---

### Decision · Program_Chairs · 2025-09-17

**Decision:**

Accept (poster)

**Comment:**

The paper proposes denoising diffusion bridge models for enabling multimodal translation. The key innovation is to use a shared latent space using a modality agnostic encoder-decoder. A combination of bridge loss, predictive loss, and multimodal contrastive loss are used to train the diffusion bridge. Experiments are provided on two  tasks: 3D shape generation from multi-view images and multi-view face generation.

The paper received overall positive reviews with the reviewers appreciating the importance of the problem setup and the technical exposition. There were important concerns raised on multiple fronts:
1) Relatively simpler benchmark tasks used in the experiments (TLPh, 8Cc5, Egf9) -- to which authors provided additional results during the rebuttal on the task of audio-to-image as well as on edges-to-bags (with aligned modalties) showing compelling results.
2) Lack of analysis on compute complexity, ablations on losses (Egf9, BFU8) -- authors point to the numerical results in the Appendix showing better run-time efficiency and model size
3) Concerns on similarities to prior works such as CrossFlow, FlowTok, DPBridge, etc. were pointed out (TLPh) -- authors argued that prior architectures are specific to certain domains, while the proposed model is modality agnostic.

AC had an independent reading of the paper and agrees with the reviewers that the paper makes an interesting contribution to the multimodal translation domain. Authors provided a strong rebuttal with additional experiments and numerical results addressing a majority of the reviewers' concerns.

Some concerns remained after the discussion, such as the need for additional ablation studies (Egf9), missing studies on non-visual domains (TLPh), and missing benchmarking studies to stronger baselines (TLPh). AC also thinks the audio-to-image results provided during the rebuttal would need visual scrutiny for quality, which is not possible.

Taking into consideration the overall reviewer sentiment and the  technical exposition, AC recommends accepting the paper. Authors should address all the remaining concerns in the camera-ready paper, including revising the paper for clarity and incorporation of the reviewers suggestions.